# Regulation of defective mitochondrial DNA accumulation and transmission in *C. elegans* by the programmed cell death and aging pathways

Sagen Flowers[1], Rushali Kothari[1], Yamila N Torres Cleuren[1,2], Melissa R Alcorn[1], Chee Kiang Ewe[1], Geneva Alok[1], Samantha L Fiallo[1], Pradeep M Joshi[1]*, Joel H Rothman[1]*

[1]Department of MCD Biology and Neuroscience Research Institute, University of California, Santa Barbara, Santa Barbara, United States; [2]Computational Biology Unit, Institute for Informatics, University of Bergen, Bergen, Norway

*For correspondence:
joshi@ucsb.edu (PMJ);
joel.rothman@lifesci.ucsb.edu
(JHR)

**Competing interest:** The authors declare that no competing interests exist.

**Abstract** The heteroplasmic state of eukaryotic cells allows for cryptic accumulation of defective mitochondrial genomes (mtDNA). 'Purifying selection' mechanisms operate to remove such dysfunctional mtDNAs. We found that activators of programmed cell death (PCD), including the CED-3 and CSP-1 caspases, the BH3-only protein CED-13, and PCD corpse engulfment factors, are required in *C. elegans* to attenuate germline abundance of a 3.1-kb mtDNA deletion mutation, *uaDf5*, which is normally stably maintained in heteroplasmy with wildtype mtDNA. In contrast, removal of CED-4/Apaf1 or a mutation in the CED-4-interacting prodomain of CED-3, do not increase accumulation of the defective mtDNA, suggesting induction of a non-canonical germline PCD mechanism or non-apoptotic action of the CED-13/caspase axis. We also found that the abundance of germline mtDNA$^{uaDf5}$ reproducibly increases with age of the mothers. This effect is transmitted to the offspring of mothers, with only partial intergenerational removal of the defective mtDNA. In mutants with elevated mtDNA$^{uaDf5}$ levels, this removal is enhanced in older mothers, suggesting an age-dependent mechanism of mtDNA quality control. Indeed, we found that both steady-state and age-dependent accumulation rates of *uaDf5* are markedly decreased in long-lived, and increased in short-lived, mutants. These findings reveal that regulators of both PCD and the aging program are required for germline mtDNA quality control and its intergenerational transmission.

## Editor's evaluation

This valuable work suggests a novel mechanism of purifying selection by which programmed cell death contributes to the selective removal of mtDNA deletion mutations in *C. elegans*. The authors demonstrate with convincing evidence that mtDNA deletion is more abundant in the offspring of older mothers and that activators for programmed cell death are required to suppress the accumulation of defective mtDNA in the germline of *C. elegans*. Because of the likely central role of mtDNA deletions in aging and age-dependent diseases, this work will be of interest to scientists in the field of mitochondrial biology as well as aging.

## Introduction

Mitochondrial diseases are a group of conditions that affect mitochondrial functions in up to 1 in 4300 people (*Ng and Turnbull, 2016*; *Chinnery, 2015*; *Gorman et al., 2015*). Generally, these

diseases present as dysfunction in the tissues or organs with the most intensive energy demands, most commonly in muscle and the nervous system (*Ghaoui and Sue, 2018*). Many of these diseases are attributable to mutations in the mitochondrial DNA (mtDNA) or nuclear DNA (nDNA), and include those disorders with defects in mitochondrial function, dynamics, or quality control, or in which there is miscommunication between the mitochondria and the endoplasmic reticulum (*Gorman et al., 2015*; *Area-Gomez and Schon, 2014*). The progressive advancement of the diseased state resulting from age-dependent accumulation of mutant mtDNA is a common trait among mitochondrial diseases (*Dhillon and Fenech, 2014*; *Park and Larsson, 2011*; *Burté et al., 2015*). While the severity of the disease varies with the nature of the mutation, the most severe phenotypes result in childhood death, as in Leigh syndrome and MELAS (*Area-Gomez and Schon, 2014*; *Schon et al., 2012*). As there are currently no pharmacological treatments for mitochondrial diseases, it is of great importance to uncover the cellular processes that underlie the regulation of mtDNA quality control.

mtDNAs show high mutation rates (*Brown et al., 1979*; *Konrad et al., 2017*; *Denver et al., 2000*) and hence it is critical that cells possess mechanisms to remove detrimental mtDNA alleles, a process called purifying selection (*Palozzi et al., 2018*; *Stewart et al., 2008*). Defects in this process can result in mitochondrial diseases, allowing harmful mtDNA mutations to persist through the maternal germline and subsequent generations. Processes that regulate mtDNA quality control include mitochondrial fission/fusion dynamics and mitophagy (*Burté et al., 2015*; *Twig and Shirihai, 2011*; *Busch et al., 2014*; *Ni et al., 2015*; *Ashrafi and Schwarz, 2013*; *Youle and Narendra, 2011*; *Tilokani et al., 2018*), and the mitochondrial unfolded protein response (UPR$^{MT}$; *Hernando-Rodríguez and Artal-Sanz, 2018*; *Rolland et al., 2019*; *Münch, 2018*; *Callegari and Dennerlein, 2018*; *Gitschlag et al., 2016*; *Nargund et al., 2012*; *Lin et al., 2016*). Further, it has also been found that the IIS pathway (*Murphy and Hu, 2013*; *Haroon et al., 2018*) ameliorates the fitness defects of mutant mtDNA.

One potential cellular process that could be used to eliminate defective mtDNAs is the culling of cells bearing mtDNA mutations by programmed cell death (PCD). The mechanisms of both developmentally controlled and genotoxicity-induced PCD have been shown to be well-conserved across metazoans (*Lord and Gunawardena, 2012*), and much of the machinery that choreographs this process is directed by mitochondria (*Jeong and Seol, 2008*; *Estaquier et al., 2012*; *Bhola and Letai, 2016*). Mitochondrial-dependent processes participating in PCD include permeabilization of the inner mitochondrial membrane and release of mitochondrial factors that mediate transduction of intermediary events in the cell suicide program (*Jeong and Seol, 2008*; *Estaquier et al., 2012*; *Bhola and Letai, 2016*). Both mitochondrial function and PCD are linked to the process of organismal aging (*Tower, 2015*). mtDNA mutations accumulate in tissues as organisms age, and it has been suggested that this accumulation is a major contributor to aging (*Larsson, 2010*; *Payne and Chinnery, 2015*; *Kauppila et al., 2017*).

The nematode *C. elegans* provides an attractive model for exploring the potential role of PCD in mtDNA purifying selection. The well-described, conserved PCD regulatory pathway in *C. elegans* functions not only to eliminate 131 somatic cells during development through a rigidly stereotyped program (*Conradt et al., 2016*), but is also activated apparently stochastically during germline development, resulting in the death of >95% of nuclei that would otherwise be destined to become oocytes in the mature hermaphrodite (*Lord and Gunawardena, 2012*; *Gumienny et al., 1999*; *Baum et al., 2005*; *Jaramillo-Lambert et al., 2007*). In addition to this 'physiological' PCD, germline nuclei that have experienced genotoxic stress are eliminated through p53-dependent apoptosis, as is also the case in somatic mammalian cells (*Hafner et al., 2019*; *Derry et al., 2001*). Thus, germline PCD allows for selective removal of nuclei with damaged genomes, thereby preventing intergenerational transmission of defective nuclear DNA. Given the prominent role played by mitochondria in the PCD process, it is conceivable that mitochondrial dysfunction could trigger PCD in the germline and, as such, might similarly provide a quality control mechanism for eliminating aberrant mtDNA, as seen for the nuclear genome.

We report here that germline mtDNA quality control in *C. elegans* is influenced by regulators of both PCD and the aging program. We find that pro-apoptotic regulators of germline PCD, notably the caspases CED-3 and CSP-1, the BH3-only domain protein CED-13, and regulators of cell corpse engulfment, reduce abundance of an mtDNA deletion and that abrogation of their functions results in elevated levels of the defective mtDNA. Notably, however, loss of the CED-3 activator CED-4/Apaf1 (*Huang et al., 2013*; *Seshagiri and Miller, 1997*) does not result in elevated levels of defective mtDNA.

These findings raise the possibilities that either the caspases and the other pro-apoptotic factors function in mtDNA purifying selection by a non-canonical CED-4-independent cell death program, or that these pro-apoptotic regulators function in mtDNA purifying selection through a PCD-independent mechanism. We also report that defective mtDNA accumulates in the germline of animals with age and that although the abundance of the defective mtDNA is reduced in offspring, progeny of older mothers inherit higher levels of the mutant mtDNA than those from young mothers. Intergenerational removal of the defective mtDNA appears to be enhanced in older animals with defective mtDNA quality control. Further, we found that lifespan-extending mutations in both the IIS pathway and the non-IIS-dependent lifespan-regulator CLK-1/MCLK1 decrease accumulation of defective mtDNA, and that short-lived mutants show elevated accumulation, implicating molecular regulators of the aging process in mtDNA purifying selection. Our findings reveal that the PCD machinery and the aging program contribute to the removal of mtDNA mutations during germline development and their intergenerational transmission.

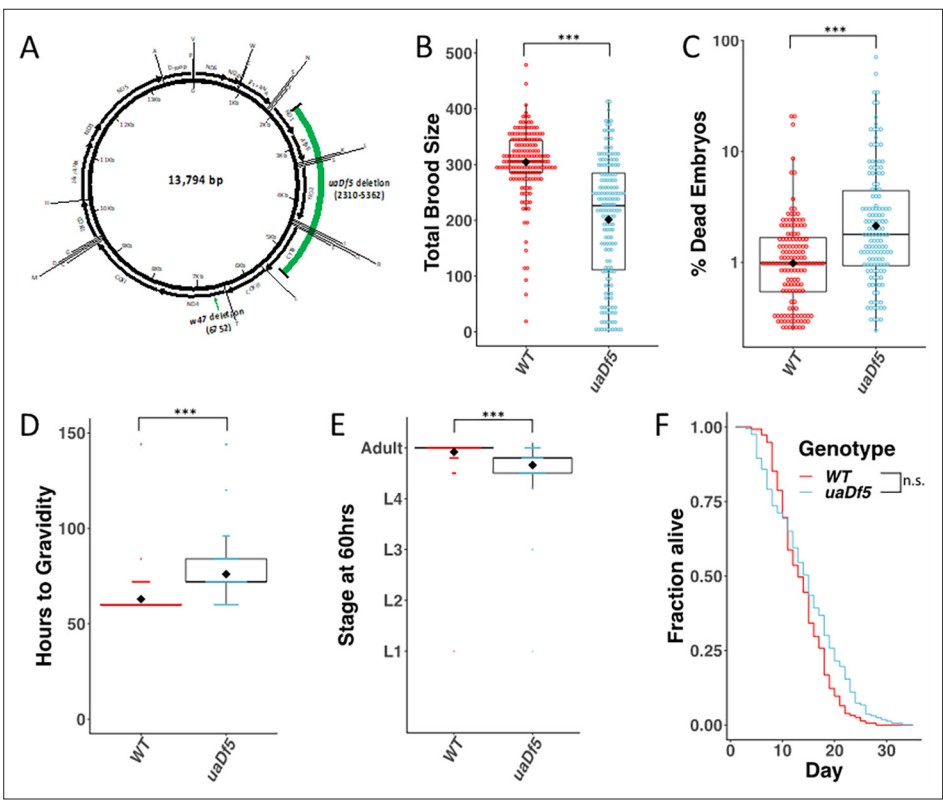

**Figure 1.** Analysis of the impact of mtDNA*uaDf5* on fitness parameters. (**A**) Diagram of *C. elegans* mtDNA. Black bars with arrows indicate the locations of genes and direction of transcription. Black lines with letters indicate the locations of tRNAs. Green bars show the locations of the mtDNA*uaDf5* deletion as well as the linked *w47* insertion that was identified via Illumina sequencing. (**B**) Brood size analysis of mtDNA*uaDf5* compared to laboratory wildtype N2. (**C**) Embryonic lethality analysis of *uaDf5* compared to laboratory wildtype N2. (**D**) Developmental rate analysis of mtDNA*uaDf5* compared to laboratory wildtype N2, counting how many hours it takes for starved L1s to reach gravidity once plated on food. (**E**) Developmental rate analysis of mtDNA*uaDf5* compared to laboratory wildtype N2, staging worms 60 hr after synchronized, starved L1s are plated on food. (**F**) Survival curve analysis of mtDNA*uaDf5* compared to laboratory wildtype N2, day 1 is defined as the day starved L1s are plated on food. Median lifespan and statistics are presented in *Figure 1—figure supplement 2*. For (**B–E**), box plots show median and IQR (Interquartile Range), and the diamond indicates the mean. Statistical analysis was performed using the Mann–Whitney test (***p < 0.001, n.s. not significant).

The online version of this article includes the following source data and figure supplement(s) for figure 1:

**Figure supplement 1.** Characterization of the *uaDf5* allele.

**Figure supplement 1—source data 1.** ddPCR reads of mitochondrial DNA.

**Figure supplement 2.** Lifespan analysis of the impact of *uaDf5*.

# Results

## The stably maintained mtDNA deletion mutant *uaDf5* contains multiple linked mutations resulting in aberrant proteins and shows deleterious effects on growth

To test the role of potential regulatory factors in mtDNA purifying selection, we took advantage of *uaDf5*, a 3.1-kb mtDNA deletion mutation that removes part or all of four protein-coding genes and seven tRNAs (*Figure 1A*; *Tsang and Lemire, 2002*). Given the presumably deleterious nature of this defective mtDNA, it was of interest to understand how it is stably transmitted despite active purifying selection processes. While its maintenance at high levels is attributable in part to stabilization by the mitochondrial UPR (*Gitschlag et al., 2016*), *uaDf5* persists, albeit at lower levels, in animals lacking this activity. One possible explanation for this phenomenon is that the mutant mtDNA might be maintained in heteroplasmy with an otherwise intact mtDNA carrying a complementing mutation. We sought to test this possibility through deep sequencing of mtDNA isolated from the *uaDf5*-bearing strain. Comprehensive sequence analysis revealed that, in addition to the large deletion, the strain indeed carries a second mtDNA mutation, *w47* (*Figure 1A*, *Figure 1—figure supplement 1A–C*). *w47* is a single base pair insertion in the *nduo-4* gene that causes a frameshift, predicted to result in a truncated NADH dehydrogenase 4 (ND4) protein lacking 321 residues (*Figure 1—figure supplement 1D*). ND4 is an essential transmembrane subunit within complex I of the mitochondrial respiratory chain (MRC), which drives NADH-oxidation-dependent transport of protons across the inner mitochondrial membrane (*Lemire, 2005*; *van der Bliek et al., 2017*; *Sousa et al., 2018*). While this raised the possibility of two complementing mtDNA genomes, we found that the *w47* mutation is present at the same abundance as the *uaDf5* deletion mutation (~75% of total mtDNA, *Figure 1—figure supplement 1B*) rather than that of the wildtype mtDNA, strongly suggesting that it resides on the same mtDNA genome. As this second mutation cannot explain stabilization of the defective mtDNA by *trans*-complementation of two deleterious mutations, other mechanisms appear to promote the stable inheritance of *uaDf5*.

In addition to the aberrant protein encoded by the *w47* frameshift mutation in *nduo-4*, a second abnormal protein is encoded by the *uaDf5* genome: one end of the deletion results in a fusion protein comprised of the first 185 amino acids of NADH dehydrogenase 1 (ND1, a homolog of the core MT-ND1 transmembrane subunit of complex I of the MRC; *Lemire, 2005*; *Sousa et al., 2018*; *Baradaran et al., 2013*), and the last 81 amino acids of mitochondrial-encoded cytochrome b (CTB-1/CYTB, a transmembrane subunit of complex III of the MRC; *Lemire, 2005*; *Sousa et al., 2018*; *Song et al., 2016*; *Figure 1—figure supplement 1E*). It is conceivable that accumulation of these two abnormal proteins – the truncated ND4 and the ND1-CYTB fusion protein resulting from *w47* and *uaDf5*, respectively – activate the UPR$^{MT}$, which has been shown to result in clearance of mtDNA$^{uaDf5}$, dependent on the ATFS-1 transcription factor (*Gitschlag et al., 2016*; *Lin et al., 2016*).

Animals harboring mtDNA$^{uaDf5}$ are viable and fertile, presumably because they contain intact wildtype mtDNA (*Tsang and Lemire, 2002*). However, we found that *uaDf5*-bearing animals displayed a significant reduction in brood size (WT 304 ± 4.8; *uaDf5* 201 ± 8.6 embryos laid, p < 0.001) (*Figure 1B*) and a significant increase in embryonic lethality (WT 1.4 ± 0.2%; *uaDf5* 4.2 ± 0.7%, p < 0.001) (*Figure 1C*). Additionally, *uaDf5* animals are slow-growing, evident in both the number of hours to reach gravidity (WT 63 ± 0.8; *uaDf5* 76 ± 1.4 hr at 20°C, p < 0.001) (*Figure 1D*) and the stage of development reached after 60 hr of feeding (WT: adult; *uaDf5:* mid-L4) (*Figure 1E*). In contrast, however, we were surprised to find that lifespan was not substantially affected (WT 14 ± 0.4; *uaDf5* 15 ± 0.5 days) (*Figure 1F*, *Figure 1—figure supplement 2*). Given the significant decline in the majority of fitness parameters tested, we conclude that *uaDf5* is a useful tool for studying mitochondrial disease and mechanisms underlying mtDNA quality control.

## PCD regulators promote removal of mtDNA$^{uaDf5}$

During germline development in *C. elegans*, as many as 95% of nuclei destined to become potential oocytes are eliminated by PCD (*Gumienny et al., 1999*; *Baum et al., 2005*; *Jaramillo-Lambert et al., 2007*; *Gartner et al., 2008*). While this process has been proposed to be stochastically determined (*Gumienny et al., 1999*; *Gartner et al., 2008*), it has also been suggested that it may function to selectively remove all but the most 'fit' germline cells. As such, PCD could perform a role in purifying

selection in the germline, wherein potential oocytes that undergo PCD are associated with higher levels of defective mtDNA. To test this hypothesis, we introduced *uaDf5* into various PCD mutants and quantified abundance of the defective mtDNA by digital-droplet PCR (ddPCR; see *Supplementary file 1* for list of mutants tested). In a wildtype genetic background, we found that the steady-state fractional abundance of *uaDf5* in populations of 200 day 1 adults (first day of adulthood) is highly reproducible across four separate trials, demonstrating the reliability and robustness of the assay. Our analyses confirmed that mtDNA$^{uaDf5}$ constitutes the major molar fraction of mtDNA in the *uaDf5*-bearing strain by a nearly 3:1 ratio (*Figure 2—figure supplement 1*).

CED-3 in *C. elegans* is the major executioner caspase in the canonical PCD pathway (*Conradt et al., 2016*; *Gartner et al., 2008*; *Cohen, 1997*) and is required for virtually all PCD both in the germline and the soma (*Figure 2—figure supplement 2* adapted from Figure 2 in *Conradt et al., 2016*). We found that two *ced-3* mutations that strongly block PCD (*Shaham et al., 1999*) showed a significant increase in the ratio of defective to normal mtDNA from a molar ratio of 2.7:1 for *ced-3(+)* to 3.4:1 for *ced-3(n717)* and 4.6:1 for *ced-3(n1286)* (*Figure 2A*). This effect is attributable to an increase in mtDNA$^{uaDf5}$ in the PCD-deficient strains and not to a decrease in wildtype mtDNA$^{WT}$. Rather, we observe an increase in the abundance of mtDNA$^{WT}$ in *ced-3(−)* mutant strains relative to *ced-3(+)* ($1.45 \times 10^5$) ranging from a statistically insignificant 1.1-fold increase ($1.67 \times 10^5$, p = 0.302) in *ced-3(n717)* to 2.2-fold increase ($3.2 \times 10^5$, p = 0.0262) in *ced-3(n1286)* (*Figure 2A*, *Supplementary file 2*). These *ced-3(−)* mutations both localize to the p15 domain of the protease portion of CED-3 (*Figure 2—figure supplement 3*), consistent with abolition of caspase activity. These findings implicate the CED-3 caspase and its p15 domain in mtDNA quality control. We found that one other mutation located in the p15 domain showed only a very slight increase that was not statistically significant (*n2454*: 2.9:1, *Figure 2B*). While it is unclear why this allele showed a weaker effect it is noteworthy that, unlike the other two mutations, which result in a dramatic alteration of the protein, this mutation is predicted to result in a relatively modest (ala → thr) single amino acid substitution.

A second caspase in *C. elegans*, CSP-1, also functions, albeit less prominently, in PCD. While loss of CSP-1 alone does not result in a strong reduction in PCD, it synergizes with loss of CED-3 both in PCD and in other caspase-dependent processes (*Figure 2—figure supplement 2*; *Denning et al., 2013*; *Shaham and Shaham, 1998*; *Jeong et al., 2020*). We found that removing CSP-1 in the *csp-1(tm917)* knockout mutant results in a significant increase in mtDNA$^{uaDf5}$ abundance to a molar ratio of 3.9:1 (*Figure 2A*, *Supplementary file 2*). Further, we found that this mutation enhances the effect of the *ced-3(n717)* mutation, increasing the mtDNA$^{uaDf5}$:mtDNA$^{WT}$ molar ratio from 3.5:1 to 4.7:1 (*Figure 2A*, *Supplementary file 2*). Together these findings demonstrate that caspase activity, and possible PCD-mediated clearance, are crucial for mtDNA quality control and function in purifying selection of defective mtDNA.

We sought to further investigate a potential role for PCD in mtDNA purifying selection by evaluating the requirement for the pro-apoptotic factor CED-13, a BH3-only domain protein that acts specifically in the germline to activate PCD (*Conradt et al., 2016*; *King et al., 2019*; *Schumacher et al., 2005*). Consistent with a requirement for PCD in purifying selection, we found that two *ced-13* alleles result in a very substantial increase in the mtDNA$^{uaDf5}$:mtDNA$^{WT}$ molar ratio (*sv32*: 4.2:1 and *tm536*: 5.1:1) (*Figure 2A*, *Supplementary file 2*), supporting the notion that CED-13 promotes removal of defective mtDNA in the germline. CED-13 functions in PCD by antagonizing the function of mitochondrially localized CED-9/Bcl-2 (*King et al., 2019*; *Schumacher et al., 2005*), which normally sequesters the apoptosome factor CED-4/APAF1 at mitochondria, thereby preventing it from triggering autocatalytic conversion of the executioner caspase zymogen proCED-3 to its pro-apoptotic protease structure (*Figure 2—figure supplement 2*; *Fairlie et al., 2006*). An equivalent action is carried out in the soma by the BH3-only protein EGL-1 (*Fairlie et al., 2006*). *n1950*, a gain-of-function allele of *ced-9* that blocks the interaction of EGL-1 with CED-9 at the mitochondria, results in elimination of PCD in the soma but not the germline (*Conradt et al., 2016*; *Gumienny et al., 1999*; *Gartner et al., 2008*; *Fairlie et al., 2006*). Consistent with the lack of effect of *ced-9(n1950gf)* on germline PCD, we found that the mtDNA$^{uaDf5}$:mtDNA$^{WT}$ molar ratio was not increased in *ced-9(n1950gf)* mutants (2.2:1) (*Figure 2B*, *Supplementary file 2*). Thus, CED-3, CSP-1, and CED-13 are required both for germline PCD and for removal of mtDNA$^{uaDf5}$.

Cells that undergo PCD are cleared by the surrounding cells in the process of engulfment and degradation, which is implemented through a set of redundant pathways that converge on the CED-10

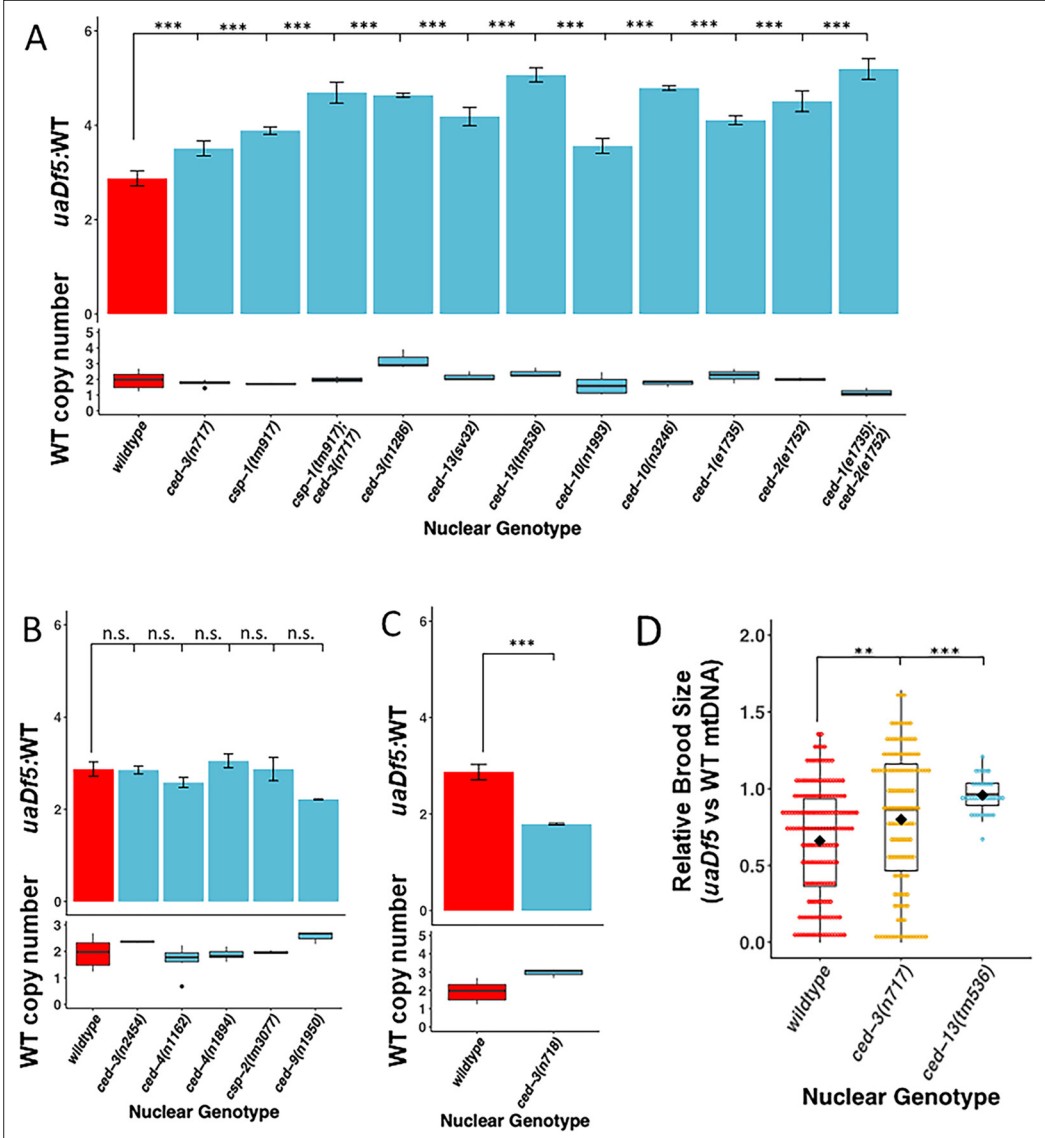

**Figure 2.** Regulators of programmed cell death (PCD) act on mutant mtDNA. (**A–C**) Digital-droplet PCR (ddPCR) analysis of the steady-state molar ratio of mtDNA$^{uaDf5}$ in 200 worm populations of day 1 adults of various PCD mutant backgrounds. (**A**) PCD mutants that result in a significant increase in the molar ratio of mtDNA$^{uaDf5}$. (**B**) PCD mutants that result in no statistical change in the molar ratio of mtDNA$^{uaDf5}$. (**C**) PCD mutant that results in a significant decrease in the molar ratio of mtDNA$^{uaDf5}$. (**D**) The relative brood size of the animals with and without mtDNA$^{uaDf5}$ in the indicated mutant backgrounds. For each nuclear genotype shown, the brood size of *uaDf5*-containing worms was normalized by dividing by the average brood size of worms containing only *WT-mtDNA*. Box plots show the median and IQR, the diamond indicates the mean. For **A–C**, *n* = 3 or more biological replicates of 200 worm populations were performed for each genotype. Average wildtype mtDNA copy number ± standard deviation is shown in the graph below in each panel. Statistical analysis was performed using one-way analysis of variance (ANOVA) with Dunnett's correction for multiple comparisons. For (**D**), statistical analysis was performed using the Mann–Whitney test. Error bars represent standard deviation of the mean (SEM) (***p < 0.001, **p < 0.01, *p < 0.05, n.s. not significant).

The online version of this article includes the following figure supplement(s) for figure 2:

**Figure supplement 1.** Reproducibility of digital-droplet PCR (ddPCR) measurement of *uaDf5*.

**Figure supplement 2.** Programmed cell death (PCD) signaling pathway.

**Figure supplement 3.** Analysis of the *ced-3* alleles.

**Figure supplement 4.** Analysis of the impact of *uaDf5* on fitness parameters in programmed cell death (PCD) mutants.

GTPase (*Wu et al., 2017*; *Kinchen et al., 2005*; *Hochreiter-hufford and Ravichandran, 2013*; *Wang and Yang, 2016*; *Figure 2—figure supplement 2*). Although this engulfment process is necessary primarily for removal of the resultant corpses, it also appears to play an active role in cell killing: inhibition of the engulfment pathway diminishes occurrence of PCD, likely through a complex feedback mechanism (*Reddien and Horvitz, 2004*; *Hoeppner et al., 2001*). Further supporting a role for PCD in purifying selection, we found that single or double mutations of several genes that promote engulfment of cell corpses result in elevated $mtDNA^{uaDf5}$:$mtDNA^{WT}$ molar ratios, ranging from 3.5:1 to 5.2:1 (4.1:1 for *ced-1(e1735)*, 4.5:1 for *ced-2(e1752)*, 5.2:1 for the *ced-1(e1735); ced-2(e1752)* double mutant, 3.6:1 for *ced-10(n1993)*, and 4.8:1 for *ced-10(n3246)*; *Figure 2A*, *Supplementary file 2*). As we observed for the caspase mutants, the increase in relative abundance of $mtDNA^{uaDf5}$ seen in the cell corpse engulfment mutants is not associated with a significant decrease in wildtype mtDNA levels.

We tested whether *generally* increased germline PCD alters $mtDNA^{uaDf5}$ abundance by examining the effect of removing the caspase-related factor, CSP-2, which has been shown to play an anti-apoptotic role through inhibition of CED-3 autoactivation in the germline (*Figure 2—figure supplement 2*; *Geng et al., 2009*). We found that a loss-of-function mutation in *csp-2*, which elevates germline PCD, did not alter the $mtDNA^{uaDf5}$:$mtDNA^{WT}$ molar ratio (2.9:1 for the *csp-2(tm3077)* knockout mutation) (*Figure 2B*). This observation does not conflict with a requirement for PCD in purifying selection: a general increase in PCD in the germline of *csp-2(−)* animals would not be expected per se to alter the mechanism that *discriminates* defective from normal mtDNAs and therefore the relative abundance of the two forms. Rather, our findings suggest that the mechanisms that recognizes and disposes of the defective mtDNA specifically requires PCD components acting in a selective, rather than general process (i.e., in those cells with the highest burden of the defective mtDNA).

As suggested above, it is conceivable that the effects described were attributable to changes in the abundance of $mtDNA^{WT}$ per se. For example, if the nuclear mutations tested resulted in decreased $mtDNA^{WT}$ copy numbers without affecting $mtDNA^{uaDf5}$ abundance, then the $mtDNA^{uaDf5}$:$mtDNA^{WT}$ molar ratios would be skewed upward. However, while we observed modest variation in $mtDNA^{WT}$ abundance across the mutant strains (*Figure 2*), across the strains, the variation does not generally correlate with the increased fractional abundance of $mtDNA^{uaDf5}$: while some mutants with elevated fractional abundance of the defective mtDNA showed somewhat higher levels of $mtDNA^{WT}$, others contained lower levels of intact mtDNA (*Figure 2*). For example, although the *ced-13(tm536)* mutant showed the highest $mtDNA^{uaDf5}$:$mtDNA^{WT}$ molar ratio, the abundance of $mtDNA^{WT}$ was also higher than in the wildtype nuclear background ($2.4 \times 10^5 \pm 3.09 \times 10^4$ in *ced-13(tm536)* vs $1.45 \times 10^5 \pm 2.44 \times 10^4$ in wildtype p = 0.026). Further, while the *ced-1(−); ced-2(−)* double mutant showed among the largest relative abundance of $mtDNA^{uaDf5}$, this strain contained the *lowest* level of $mtDNA^{WT}$ ($1.15 \times 10^5 \pm 3.02 \times 10^3$, p = 0.254). Thus, increased levels of defective mtDNA in the PCD mutants are not apparently attributable to alterations in $mtDNA^{WT}$ levels.

## Evidence for a non-canonical PCD pathway in mtDNA purifying selection

The foregoing results implicate a role for the pro-apoptotic CED-3 and CSP-1 caspases, CED-13, and the CED-1, 2, and -10 cell corpse engulfment factors in mtDNA purifying selection. However, several observations suggest that mitochondrial purifying selection may be regulated by a non-canonical cell death pathway, in contrast to the pathway that regulates normal, physiological germline PCD.

First, we found that the *ced-3(n718)* allele lowers, rather than elevates, the abundance of $mtDNA^{uaDf5}$ ($mtDNA^{uaDf5}$:$mtDNA^{WT}$ molar ratio of 1.8:1, *Figure 2C*). This effect is likely to be attributable to the nature of the *n718* mutation. The *ced-3* mutations that result in increased $mtDNA^{uaDf5}$ levels (*Figure 2A*) alter the p15 domain, which is essential for active caspase function. In contrast, the *n718* mutation changes a residue in the caspase activation and recruitment domain (CARD), located within the prodomain of the CED-3 zymogen, which is removed upon caspase activation and affects its activation by CED-4 (*Figure 2—figure supplement 3*; *Huang et al., 2013*; *Shaham et al., 1999*). While *ced-3(n718)* strongly compromises PCD, this mutation might not alter CED-3 caspase function in a way that interferes with its role in mtDNA purifying selection.

Our surprising finding that while CED-3 activity is required for mtDNA purifying selection, a CED-3 mutation that compromises its activation by CED-4 did not elevate $mtDNA^{uaDf5}$ levels prompted us to investigate the requirement of CED-4 in mitochondrial purifying selection. Consistent with the effect

of the *ced-3(n718)* mutation, we found that eliminating the function of the pro-apoptotic regulator CED-4, the *C. elegans* ortholog of mammalian Apaf1 and the upstream activator of CED-3 in the canonical PCD pathway (*Fairlie et al., 2006*; *Yang et al., 1998*), did not result in a marked increase in the relative abundance of mtDNA$^{uaDf5}$. That is, while the mtDNA$^{uaDf5}$:mtDNA$^{WT}$ molar ratio increased to 3.1:1 in the *ced-4(n1894)* mutant, the effect was not statistically significant. Moreover, the canonical allele *ced-4(n1162)* allele similarly showed no elevation in mtDNA$^{uaDf5}$ (molar ratio = 2.6:1; *Figure 2B*). These results suggest that CED-3 caspase functions in mitochondrial purifying selection independently of the caspase-activating factor CED-4.

## Evidence that decreased fitness, but not lifespan, is attributable to mtDNA$^{uaDf5}$-induced PCD

Taken together, our results implicate many PCD regulatory factors, and potentially PCD, in the selective clearance of defective germline mtDNAs. Our additional observations suggest that defective mtDNAs may, in fact, *trigger* elevated germline PCD, resulting in the production of fewer mature gametes and progeny. Specifically, we found that the significant decrease in brood size that we observed in *uaDf5*-bearing animals with a wildtype nuclear background is partially suppressed by both *ced-3(−)* and *ced-13(−)* mutations, which prevent PCD (*Figure 2D*, *Figure 2—figure supplement 4A*), suggesting that elimination of PCD might rescue cells that would otherwise be fated to die as a result of accumulation of defective mtDNA. Our findings further underscore the observation that accumulation of defective mtDNA in those animals that do survive does not affect longevity, as we found that lifespan is unaltered in these PCD mutants even when the levels of mtDNA$^{uaDf5}$ are nearly doubled (*Figure 2—figure supplement 4B, C*).

## Age-dependent accumulation of mtDNA$^{uaDf5}$ in the germline

Our findings that PCD regulators are required to reduce mtDNA$^{uaDf5}$ abundance, the central role that mitochondria play in PCD (*Jeong and Seol, 2008*; *Estaquier et al., 2012*; *Bhola and Letai, 2016*; *Parsons and Green, 2010*), the observed decline of mitochondrial health during the aging process (*Park and Larsson, 2011*; *Larsson, 2010*; *Payne and Chinnery, 2015*; *Kauppila et al., 2017*; *Harman, 1992*; *Harman, 1956*; *Szczepanowska and Trifunovic, 2017*; *Bratic and Larsson, 2013*; *Ziegler et al., 2015*), and the relationship between excessive PCD and the aging phenotype (*Tower, 2015*) led us to examine the dynamics of mtDNA$^{uaDf5}$ accumulation as worms age. We measured the fractional abundance of *uaDf5* in adults at progressively increased ages spanning day 1, defined as the first day of egg-laying, through day 10. Day 1 through day 4 of adulthood encompasses the time during which nearly all self-progeny are produced. After day 4, hermaphrodite sperm become depleted and the animals transition into a post-gravid, progressively aging state (*Kimble and Crittenden, 2005*; *Yoon et al., 2017*; *Angeles-Albores et al., 2017*). By day 10, animals exhibit indications of advanced age. Analysis of the abundance of mtDNA$^{uaDf5}$ revealed a progressive increase throughout gravidity and post-reproductive aging, with the mtDNA$^{uaDf5}$:mtDNA$^{WT}$ molar ratio increasing from 2.9:1 to 5.5:1 (*Figure 3A*, *Figure 3—figure supplement 1A*). This age-related accumulation of *uaDf5* in adult worms is reminiscent of the accumulation of mtDNA mutations seen in aging mammals (*Larsson, 2010*; *Szczepanowska and Trifunovic, 2017*; *Kujoth et al., 2005*) and suggests that *uaDf5* in *C. elegans* may be a useful tool for studying the role that mtDNA mutations play in aging.

Given that gametes are depleted with age, it is conceivable that the age-dependent increase in mtDNA$^{uaDf5}$ is attributable to accumulation in somatic mitochondria. To assess whether the observed age-related accumulation of mtDNA$^{uaDf5}$ occurs predominantly in the maternal germline or in somatic cells, we analyzed animals defective in germline development by taking advantage of the *glp-4(bn2)* mutant, which produces only a small number (~12) of germline cells compared to that in wildtype animals (~1500), with no known effect on somatic gonad development (*Beanan and Strome, 1992*). In contrast to the increased mtDNA$^{uaDf5}$ abundance with age seen at permissive temperature (mtDNA$^{uaDf5}$:mtDNA$^{WT}$ of 1.8:1 at day 1, rising to 2.6:1 at day 4, for an overall increase by 48%), we found that *glp-4(bn2)* animals at the non-permissive temperature showed a slight decrease in the defective mtDNA from day 1 to 4 of adulthood (2.2:1 at day 1, dropping to 2:1 at day 4, for an overall decrease of 6.3%) (*Figure 3B*). These results strongly suggest that the observed age-dependent increase in mtDNA$^{uaDf5}$ abundance occurs exclusively in the germline. We found that mtDNA$^{uaDf5}$ does eventually appear to accumulate in somatic cells with age, as day 10 adults grown at the restrictive temperature

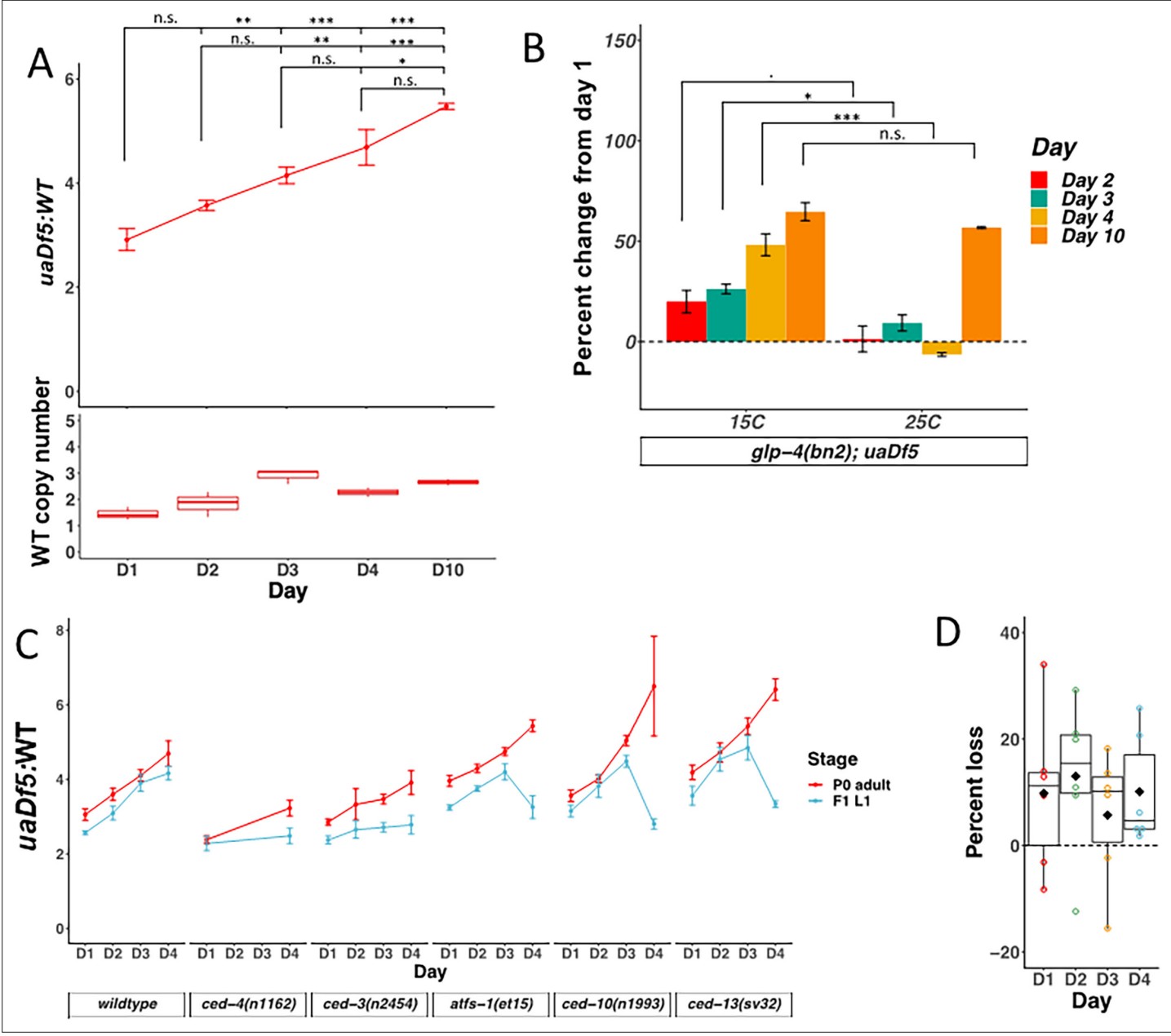

**Figure 3.** mtDNA*uaDf5* accumulates in the germline of aging adults, and evidence of purifying selection between mother and offspring. (**A**) Analysis of the molar ratio of mtDNA*uaDf5* in aging adults in a wildtype nuclear background. Average wildtype mtDNA copy number ± standard deviation is shown in the graph below. (**B**) Analysis of the percent change of *uaDf5*:WT from day 1 (*Y* axis = (*uaDf5*:WT day x − *uaDf5*:WT day 1)/(*uaDf5*:WT day 1)). For *glp-1(q231ts)*, *fem-3(q20ts)*, and *glp-4(bn2ts)*, 15°C is the permissive temperature (germline development occurs) and 25°C is the restrictive temperature (female germline development is inhibited). Statistical analysis was performed using one-way analysis of variance (ANOVA) with Tukey correction for multiple comparisons (***p < 0.001, **p < 0.01, *p < 0.05, n.s., not significant). Error bars represent standard deviation of the mean (SEM). (**C**) Analysis of the molar ratio of mtDNA*uaDf5* in aging adults (P0 adult) and their L1 progeny (F1-L1) in various nuclear backgrounds shows that all strains decrease the *uaDf5* load during transmission from mother to offspring, and that strains with significantly higher mtDNA*uaDf5* levels (*atfs-1(et15)*, *ced-10(n1993)*, and *ced-13(sv32)*) have a more significant removal mechanism at day 4 of adulthood. *n* = 3 or more replicates of 200 worm populations were performed for each timepoint. Error bars represent SEM. Gray dashed line indicates a hypothetical threshold at which high mtDNA*uaDf5* burden activates enhanced intergenerational purifying selection in older mothers. (**D**) Analysis of the measured loss of mtDNA*uaDf5* between mother and offspring at each day of adulthood shows that mtDNA*uaDf5* removal occurs. *n* = 6 replicates of 200 worm populations for each condition.

The online version of this article includes the following figure supplement(s) for figure 3:

**Figure supplement 1.** *uaDf5* accumulation in individual lines of adults and progeny.

showed a marked increase in the mtDNA$^{uaDf5}$:mtDNA$^{WT}$ molar ratio compared to day 1 adults even in the absence of a germline (day 10 *glp-4(bn2)* molar ratio of 3.4:1, a 57% increase from day 1 levels) (**Figure 3B**). We also observed an increase in wildtype mtDNA copy number during the period of gravidity, which is likely the result of compensatory expansion in response to the replicative advantage of mtDNA$^{uaDf5}$. We conclude that the marked increase in mtDNA$^{uaDf5}$ with age during the period of fecundity occurs primarily in the germline and that the defective mtDNA accumulates in both germline and somatic tissue during post-reproductive life.

## Age-dependent increase in mtDNA$^{uaDf5}$ burden is transmitted to progeny

As the mtDNA is inherited strictly through the maternal germline, we posited that the age-dependent increase in the fractional abundance of germline mtDNA$^{uaDf5}$ might be transmitted to progeny animals. To test this hypothesis, we measured the molar ratio of mtDNA$^{uaDf5}$ in 200-worm populations of L1 larvae derived from day 1 to 4 adults (**Figure 3C**, **Figure 3—figure supplement 1B**). This analysis led to two key observations: (1) the abundance of mtDNA$^{uaDf5}$ is reduced during transmission between mother and offspring (average decrease ranging from 6% to 13%), presumably as a result of purifying selection, and (2) the abundance of the defective mtDNA in the offspring correlates with the age of the mothers: the progeny of older mothers contain a markedly higher mtDNA$^{uaDf5}$:mtDNA$^{WT}$ molar ratio (4.2:1) than that of younger mothers (2.6:1) (**Figure 3C, D**). A similar trend was observed for mother-to-offspring transmission in five mutant strains with altered levels of mtDNA$^{uaDf5}$ (see below): in all cases, progeny contain lower abundance of mtDNA$^{uaDf5}$ than their mothers, and progeny of younger adults inherit a lower load of mtDNA$^{uaDf5}$ than progeny of older adults (**Figure 3C**). These results reveal that mtDNA quality control occurs between primordial germ cell proliferation in the female germline and L1 hatching, that is, during oocyte maturation, embryogenesis, or both.

## The lifespan-determining IIS pathway regulates accumulation of mtDNA$^{uaDf5}$

We sought to determine whether the age-dependent accumulation of defective mtDNA is controlled by known molecular mechanisms that drive the aging program in *C. elegans*. The most prominent of these regulatory systems is the highly conserved insulin/IGF-1 (insulin-like growth factor-1) pathway (IIS), which performs a pivotal regulatory function in aging and longevity (**Murphy and Hu, 2013**; **Bartke, 2008**; **Anisimov and Bartke, 2013**; **Figure 4—figure supplement 1**). Abrogation of the IIS signaling pathways, for example, as a result of mutations in the gene encoding the IIS receptor (DAF-2, in *C. elegans*), results in marked slowing of the aging program and extension of lifespan in worms, flies, and mice (**Piper et al., 2008**; **Kenyon et al., 1993**; **Tatar et al., 2001**; **Blüher et al., 2003**). The IIS pathway also functions in a broad set of other processes including, in *C. elegans*, activation of two stages of developmental arrest, or diapause, at the L1 larval stage and in formation of the dispersal form, the dauer larva, as well as in the control of germline proliferation, stress resistance, fat metabolism, and neuronal/behavioral programs (**Murphy and Hu, 2013**). It was also reported that inhibition of the IIS pathway rescues various fitness parameters in a mtDNA mutator strain which contains a faulty mtDNA polymerase (**Haroon et al., 2018**), consistent with a possible role in mtDNA quality control.

We found that two mutant alleles of *daf-2* that affect the kinase domain and which reduce rates of aging and increase lifespan, result in dramatically decreased mtDNA$^{uaDf5}$:mtDNA$^{WT}$ molar ratios from 2.8:1 to as low as 0.3:1 (0.3:1 for *daf-2(e1391)*; 0.8:1 for *daf-2(e1370)*; **Figure 4A**; see **Supplementary file 3** for a list of lifespan mutants used in the analyses). Thus, the lifespan-extending effects of *daf-2* mutations are strongly correlated with diminished abundance of defective mtDNA, to the extent that it becomes the minor species of mtDNA.

The DAF-2 receptor acts by antagonizing the DAF-16/FoxO transcription factor, the major effector of IIS downstream, in response to insulin-like ligands. Thus, removal of *daf-16* function reverses the lifespan-extending effects of *daf-2(−)* mutants. We tested whether the DAF-2 → DAF-16 pathway similarly functions in mtDNA purifying selection. We found that eliminating DAF-16 in two *daf-16* mutants results in slightly increased, albeit not statistically significant, mtDNA$^{uaDf5}$:mtDNA$^{WT}$ molar ratios (3.3:1 for *daf-16(mu86)*; 3.5:1 for *daf-16(mgDf50)*) (**Figure 4B**). Further, we found that removal of DAF-16 in the *daf-16(mu86)* mutant suppressed the decreased mtDNA$^{uaDf5}$:mtDNA$^{WT}$ molar ratios

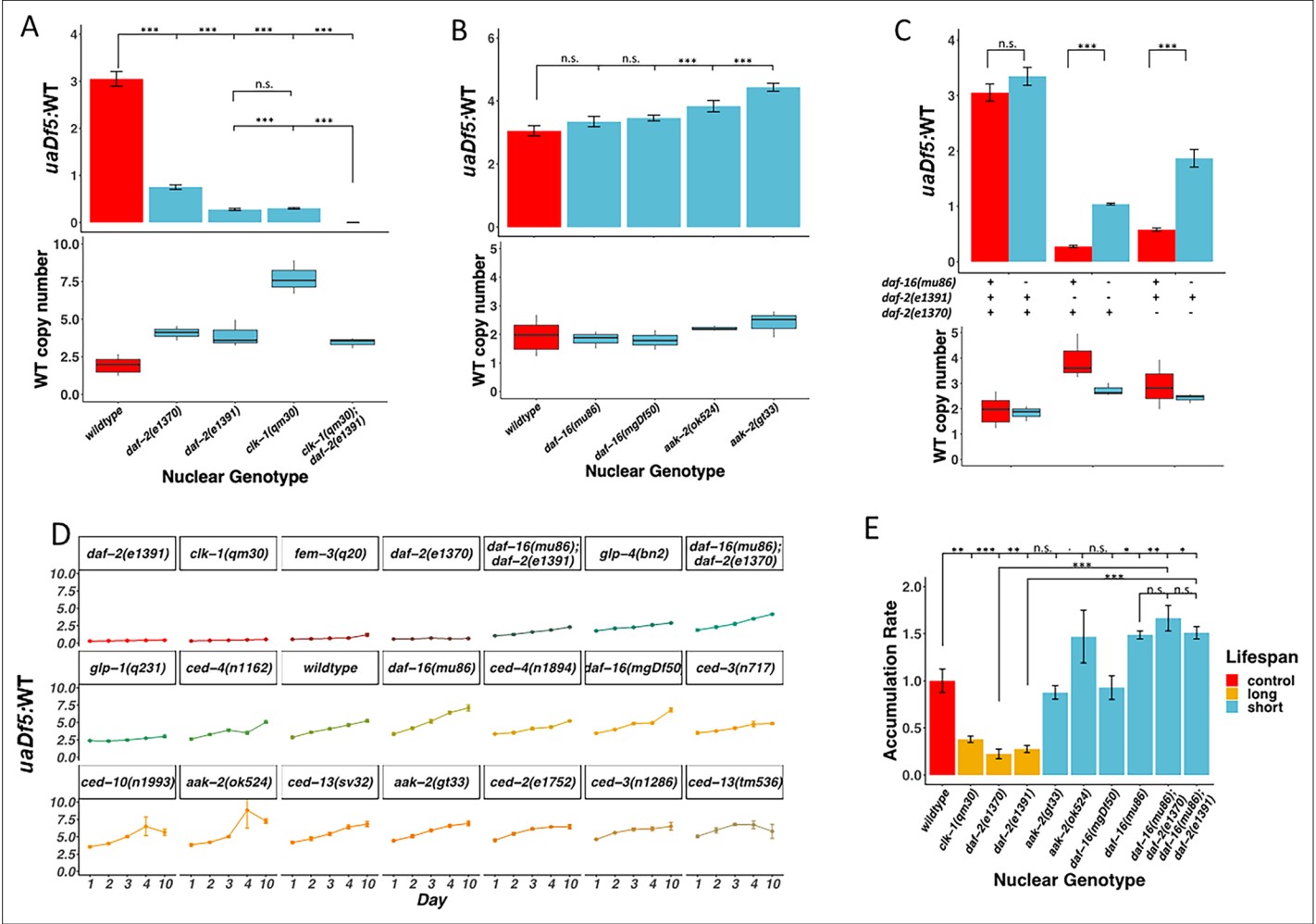

**Figure 4.** Lifespan mutants have both a lower steady-state level and accumulation rate of mtDNA$^{uaDf5}$. (**A–C**) Analysis of molar ratio of mtDNA$^{uaDf5}$ in day 1 adults of various mutant backgrounds. Average wildtype mtDNA copy number ± standard deviation is shown in the graph below in each panel. (**A**) Analysis of steady-state mtDNA$^{uaDf5}$ levels in long-lived mutants. (**B**) Analysis of steady-state mtDNA$^{uaDf5}$ levels in short-lived mutants, showing synergistic activity on mtDNA$^{uaDf5}$ removal capacity in the *daf-2(e1391) clk-1(qm30)* double mutant. (**C**) Analysis of steady-state mtDNA$^{uaDf5}$ levels in *daf-2(–)* single and *daf-16(–); daf-2(–)* double mutants, showing a partial rescue of *daf-2(–)* phenotype by *daf-16(–)*. (**D**) Analysis of the molar ratio of mtDNA$^{uaDf5}$ in aging adults in 21 different nuclear backgrounds shows a consistent accumulation trend. (**E**) Summary of the rate of increase for the lifespan regulation mutants, showing that *daf-16* rescues the *daf-2* accumulation rate phenotype. The normalized accumulation rate was calculated by fitting a regression line for each trial and then dividing the slope of the regression line by the slope of the averaged regression line found in a wildtype background. For all, n = 3 or more replicates of 200 worm populations for each genotype and stage. For **A and E**, statistical analysis was performed using one-way analysis of variance (ANOVA) with Tukey correction for multiple comparisons. For **C**, statistical analysis was performed using one-way ANOVA. For **B**, statistical analysis was performed using one-way ANOVA with Dunnett's correction for multiple comparisons. Error bars represent standard deviation of the mean (SEM) (***p <0 .001, **p <0 .01, *p < 0.05, n.s., not significant).

The online version of this article includes the following figure supplement(s) for figure 4:

**Figure supplement 1.** Insulin/IGF-1 signaling (IIS) pathway.

**Figure supplement 2.** Analysis of the accumulation of mtDNA$^{uaDf5}$ in programmed cell death (PCD) and lifespan mutants.

**Figure supplement 3.** *uaDf5* differentially impacts fitness parameters in lifespan-affecting mutants.

observed in two *daf-2* mutants, from 0.3:1 for *daf-2(e1391)* to 1:1 for *daf-16(mu86); daf-2(e1391)* and from 0.6:1 for *daf-2(e1370)* to 1.8:1 for *daf-16(mu86); daf-2(e1370)*, consistent with observations reported in a recent study (*Kirby and Patel, 2021*). While the *daf-2(–)* effect on mtDNA$^{uaDf5}$ levels is largely dependent on DAF-16, neither double mutant restored mtDNA$^{uaDf5}$ levels to those seen in animals with a fully intact IIS pathway, suggesting that other DAF-2 targets might participate in removal of defective mtDNA (*Figure 4C*). We found, conversely, that two mutations that reduce

lifespan by eliminating the function of AAK-2 (AMP activated kinase-2), a conserved factor acting in the IIS pathway (*Murphy and Hu, 2013*; *Haroon et al., 2018*), result in elevated mtDNA$^{uaDf5}$:mtDNA$^{WT}$ molar ratios as high as 4.4:1 compared to 2.8:1 in wildtype animals (3.8:1 for *aak-2(ok524)* and 4.4:1 for *aak-2(gt33)*) (*Figure 4B*). These findings demonstrate that alterations in the IIS pathway coordinately affect both lifespan and accumulation of defective mtDNA and that the DAF-2/DAF-16/AAK-2 axis acts similarly in both processes.

## Synergistic effect of multiple aging pathways on mtDNA$^{uaDf5}$ accumulation

In addition to IIS, other molecular regulatory pathways independently contribute to the rate of aging. These include CLK-1, a mitochondrial hydroxylase that functions in the pathway for ubiquinone synthesis (*Miyadera et al., 2002*; *Miyadera et al., 2001*). *clk-1* mutants with a wildtype mitochondrial genome have been shown to contain levels of mtDNA that are elevated by 30%, perhaps as the result of a compensatory process that increases demand on mitochondrial abundance, or the action of CLK-1 as a regulator of mtDNA abundance in response to energy availability within the cell (*Kirby and Patel, 2021*). As with long-lived *daf-2* mutants, we found that long-lived *clk-1(qm30)* mutants showed a greatly diminished mtDNA$^{uaDf5}$:mtDNA$^{WT}$ molar ratio of 0.3:1 (*Figure 4A*), comparable to that in *daf-2* mutants; again, mtDNA$^{uaDf5}$ is the minor species in these animals. As the IIS pathway and CLK-1 appear to act separately in controlling lifespan, we postulated that elimination of both mechanisms might further reduce levels of the defective mtDNA. Indeed, we found that mtDNA$^{uaDf5}$ was completely eliminated in *daf-2(e1391) clk-1(qm30)* double mutants, revealing a strongly synergistic effect between the two age-determining systems (*Figure 4A*). Thus, distinct regulatory pathways for longevity modulate the abundance of mtDNA$^{uaDf5}$ by apparently different mechanisms and elimination of the two pathways abrogates its maintenance.

## Age-dependent accumulation rate of mtDNA$^{uaDf5}$ strongly correlates with genetically altered rates of aging

Our findings that the steady-state abundance of mtDNA$^{uaDf5}$ increases with maternal age and that mutants with increased lifespan show lower levels of the defective mtDNA raised the possibility that purifying selection is subject to the same control as aging clocks. To assess this potential connection, we analyzed the time-dependent rates of mtDNA$^{uaDf5}$ accumulation in animals with genetic backgrounds that alter the aging clock. Analysis of 21 different genetic backgrounds over the first 10 days of adulthood revealed that the age-dependent progressive accumulation of mtDNA$^{uaDf5}$ is a consistent phenomenon (*Figure 4D*). Comparison of long-lived mutants and wildtype using a linear regression model revealed a striking positive correlation between aging rate and age-dependent rate of accumulation of mtDNA$^{uaDf5}$: all long-lived mutants in either the IIS pathway or *clk-1* accumulate mtDNA$^{uaDf5}$ at a substantially slower rate as they age than do wildtype animals (*Figure 4E*, *Figure 4—figure supplement 2A, B*). Conversely, we analyzed six short-lived IIS pathway mutant combinations and found that the *aak-2(ok524)* and *daf-16(mu86)* single mutants and *daf-16(−);daf-2(−)* double mutants all showed increased rates of mtDNA$^{uaDf5}$ accumulation (*Figure 4E*, *Figure 4—figure supplement 2C, D*). These observations suggest that in both slower- and faster-aging strains, the rate of accumulation of deleterious mtDNA is a predictor of aging rate. In the two exceptional cases, the *aak-2(gt33)* and *daf-16(mgDf50)* single mutants, we did not observe an increased accumulation rate compared to wildtype; however, the mtDNA$^{uaDf5}$ levels are consistently higher than in these two mutants than in wildtype at all stages (*Figure 4E*, *Figure 4—figure supplement 2D*) and thus diminished removal of mtDNA$^{uaDf5}$ overall correlates with decreased lifespan in these mutants as well. The greater mtDNA$^{uaDf5}$ accumulation rates seen in the short-lived animals is not attributable to the higher steady-state levels per se, as the rates of accumulation of defective mtDNA observed in the PCD mutants with higher mtDNA$^{uaDf5}$ levels show no correlation with the steady-state levels (*Figure 4—figure supplement 2E–H*); rather these increased rates appear specifically to be a property of the shortened lifespan mutants.

Consistent with a relationship between aging rates and accumulation of defective mtDNA, we found that the brood size is decreased and embryonic lethality is increased in short-lived *daf-16(−)* mutants, but not in the long-lived *clk-1(−)* mutant compared to those in a wildtype nuclear background

(*Figure 4—figure supplement 3*). These results are consistent with the possibility that longevity pathways modulate fitness in part by regulating mitochondrial homeostasis.

## Evidence for late adulthood-specific mechanisms for removal of mtDNA$^{uaDf5}$

We obtained evidence that defective mtDNA is more effectively removed in offspring of aging adults that carry an unusually high burden of mtDNA$^{uaDf5}$. The offspring of day 4 adults in those strains ('high' strains) with significantly higher steady-state fractional abundance of mtDNA$^{uaDf5}$ showed significantly greater rates of reduction of the defective mtDNA ranging from 24% reduction in *ced-10(n1993)* (fractional abundance of mtDNA$^{uaDf5}$ of 6.5:1–2.8:1) and 17.4% in *ced-13(sv32)* (6.4:1–3.3:1) to 14.5% in *atfs-1(et15)* (5.4:1–3.3:1), compared to offspring of ('low' strain) mothers with lower steady-state fractional abundance of the mutant mtDNA (13.5% reduction in *ced-3(n2454)* (3.9:1–2.8:1); 12.7% in *ced-4(n1162)* (3.2:1–2.5:1) versus only 3.2% in WT (4.7:1–4.2:1)). Remarkably, therefore, day 4 progeny from 'high' strains actually inherit a *lower* mtDNA$^{uaDf5}$ load than their siblings born from day 1 to 3 mothers (*Figures 3C and 5A*). Indeed, we found a strong correlation ($r^2$ = 0.61, p < 0.001) between the steady-state level of mtDNA$^{uaDf5}$ in mothers and the capacity for its removal between mother and progeny during day 4 of adulthood (*Figure 5B*). These results raise the possibility that very high levels of mtDNA$^{uaDf5}$ in older mothers activate an additional mtDNA purifying selection process such as mitophagy, perhaps independent of the UPR$^{MT}$ and PCD machinery, thereby ensuring that progeny are not overloaded with defective mitochondria.

## Discussion

We have obtained several lines of evidence indicating that regulators of PCD and the aging program function in mtDNA quality control and accumulation of defective mtDNA in *C. elegans*. We report eight major findings: (1) regulators of germline PCD are required for effective removal of deleterious mtDNA from the germline; (2) the cell death machinery functions in a non-canonical caspase-dependent but CED-4-independent cell death pathway to mediate mitochondrial purifying selection; (3) the CSP-1 caspase has as strong of an effect on mitochondrial purifying selection as the major PCD regulator CED-3; (4) mtDNA$^{uaDf5}$ progressively accumulates in the germline as adults age; (5) this age-dependent accumulation of mtDNA$^{uaDf5}$ is transmitted to progeny; however, the burden of the defective mtDNA is lower in offspring than mothers suggesting intergenerational purifying selection; (6) two separate aging pathways, the IIS and CLK-1 pathways, act synergistically to regulate mtDNA$^{uaDf5}$ levels; longer-lived mutants show reduced levels of the defective mtDNA while shorter-lived mutants show increased levels compared to otherwise wildtype animals; (7) the rate of mtDNA$^{uaDf5}$ accumulation is inversely correlated with lifespan in aging mutants; (8) intergenerational removal of mtDNA$^{uaDf5}$ occurs more effectively during transmission from older mothers with high burden of the defective mtDNA.

Previous reports demonstrated that UPR$^{MT}$ limits *uaDf5* clearance, and that eliminating the UPR$^{MT}$-mediating transcription factor, ATFS-1, lowers mtDNA$^{uaDf5}$ abundance (*Gitschlag et al., 2016*; *Lin et al., 2016*). Our identification of a second mutation in the *uaDf5* mutant that results in premature truncation of the ND4 gene product raises the possibility that expression of both the truncated ND4 and the ND1-CYTB fusion protein might together activate UPR$^{MT}$. The possibility that production of aberrant polypeptides resulting from these mutations that trigger this response will require analysis of additional mtDNA mutants. It has also been demonstrated that mtDNA$^{uaDf5}$ behaves as a selfish genetic element (*Gitschlag et al., 2016*) that exhibits a replicative advantage over wildtype mtDNA. To meet the metabolic energy demands of a cell, an optimal level of mtDNA is presumably maintained by regulating the mtDNA copy number. When the wildtype mtDNA copy number is insufficient, it seems likely that replication is induced and replication of the smaller deleted mtDNA$^{uaDf5}$ is amplified as a consequence. Interestingly, we found that an increase in mtDNA$^{uaDf5}$ levels, either with age or in particular mutant backgrounds, was associated with a slight increase in wildtype mtDNA levels as well, reflecting such a potential compensatory mechanism.

## Mitochondrial deletion mutant *uaDf5* as a model for mitochondrial disease

We found that *uaDf5* affects brood size, embryonic lethality, and developmental rate, highlighting its use as a model for investigating mitochondrial diseases. The reduced brood size in *uaDf5*-bearing

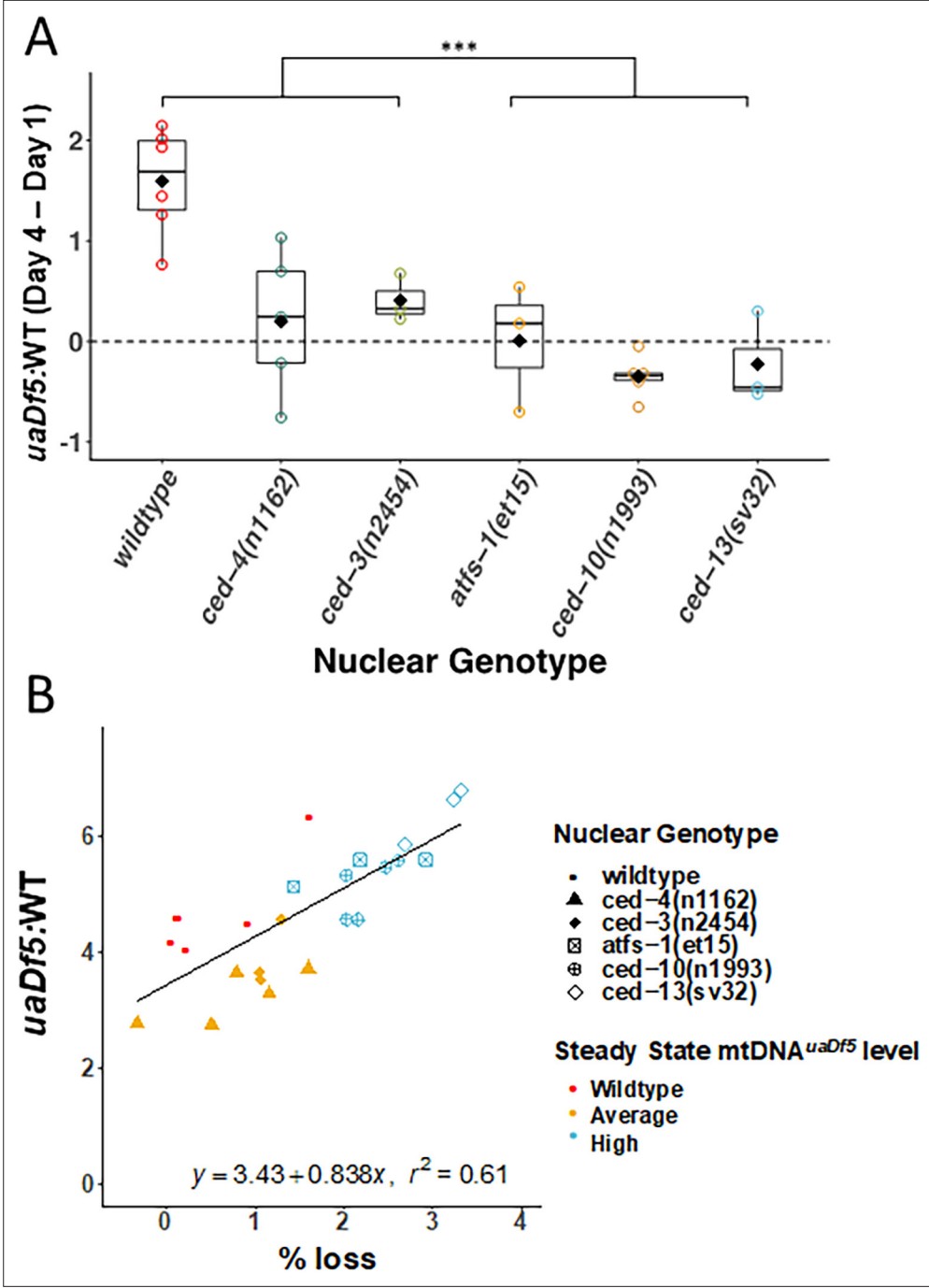

**Figure 5.** Evidence for late adulthood-specific mechanisms for removal of mtDNA$^{uaDf5}$. (**A**) Subtracting $uaDf5$:WT in progeny from day 1 adults from progeny of day 4 adults shows that day 4 F1-L1s tend to have higher mtDNA$^{uaDf5}$ burden than their day 1 siblings, but this is no longer the case in nuclear backgrounds that result in a significantly higher steady-state levels of mtDNA$^{uaDf5}$ in the adult. $n$ = 3 or more replicates for each genotype and statistical analysis was performed using the Mann–Whitney test. (**B**) Comparison of the molar ratio of mtDNA$^{uaDf5}$ in day 4 adult mothers to the absolute % removal of mtDNA$^{uaDf5}$ from mother to offspring shows a positive correlation (***p < 0.001).

animals might reflect diminished germ cell proliferation, as mitochondria have been implicated in progression of germline maturation (*Charmpilas and Tavernarakis, 2020*; *Folmes et al., 2016*). Alternatively, the defective mtDNA might trigger hyperactivation of the germline PCD pathway that specifically removes germ cells with the highest burden of defective mtDNA, as suggested by our results,

resulting in the survival of fewer mature oocytes. The increased embryonic lethality in the *uaDf5* strain may be a consequence of a genetic bottleneck effect, leading to rapid differences in mtDNA allele frequencies (*Floros et al., 2018*; *Zhang et al., 2018*; *Wai et al., 2008*) and a subpopulation of oocytes containing levels of mtDNA$^{uaDf5}$ that exceed a threshold required for viability.

We were surprised to find, in contrast to a previous report (*Liau et al., 2007*), that mtDNA$^{uaDf5}$ did not alter lifespan. Given that mitochondrial mutations are often coupled with compensatory mutations in the nuclear genome (*Zhu et al., 2019*; *Levin et al., 2014*; *Paliwal et al., 2014*; *Zhu et al., 2014*; *Meiklejohn et al., 2013*), one possible explanation for this discrepancy might be that a compensatory nuclear mutation exists in the strain analyzed, diminishing the impact of the defective mitochondrial genome. We note, however, that we backcrossed the *uaDf5* strain extensively to the laboratory reference strain N2 prior to performing the reported analyses. It is conceivable that although we observed

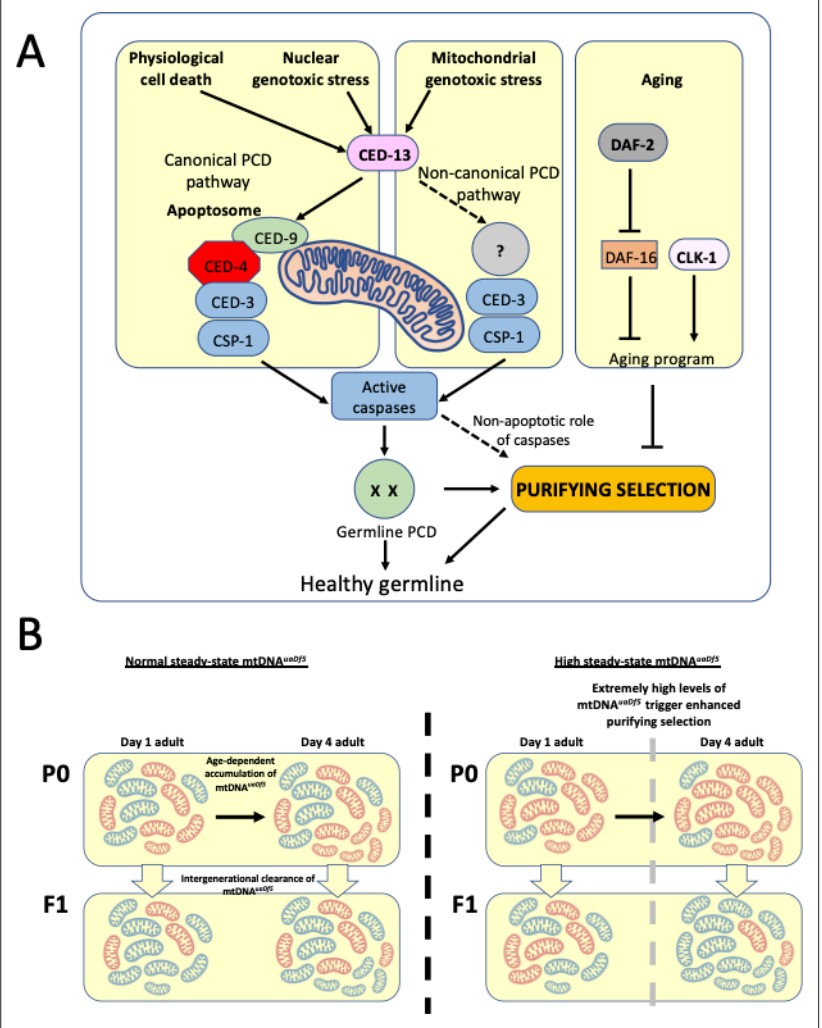

**Figure 6.** Regulation of mtDNA$^{uaDf5}$ accumulation and transmission by the programmed cell death (PCD) and aging pathways. (**A**) Our results suggest that CED-3 and CSP-1, which are activated by BH-3 only protein CED-13, function cooperatively to promote mitochondrial purifying selection, independent of CED-9 and the CED-4 apoptosome. The clearance of mtDNA$^{uaDf5}$ may therefore involve induction of a non-canonical germline PCD mechanism or non-apoptotic action of the CED-13/caspase axis. Additionally, Insulin/IGF-1 signaling (IIS) and CLK-1 aging pathways act synergistically to regulate mitochondrial purifying selection. (**B**) mtDNA$^{uaDf5}$ (red) accumulates in the germline relative to mtDNA$^{WT}$ (blue) as adults age and the increased mtDNA$^{uaDf5}$ levels are transmitted to the progeny, although the mtDNA$^{uaDf5}$ burden is consistently lower in progeny than mothers. This intergenerational purifying selection is enhanced in the older mothers of mutants with high steady-state mtDNA$^{uaDf5}$ (e.g., *ced-13*, *ced-10*, and *atfs-1*), suggesting a threshold beyond which a germline PCD-independent mtDNA quality control process may be initiated or enhanced in these older mothers.

no effect in the lab, *uaDf5* might alter lifespan under natural conditions. Exposure to increased stress from growth in the wild might be less tolerated in animals bearing mtDNA$^{uaDf5}$, as has been observed with other mitochondrial mutants (*Dingley et al., 2014*), resulting in diminished lifespan.

## Caspases and cell death machinery regulate mitochondrial purifying selection

Our results lend support to the hypothesis that germline PCD mechanisms may be used to cull germline cells with defective mtDNA (*Figure 6A*). Developmentally PCD and cell death in response to genotoxic stress are mediated by caspases upon activation by the *C. elegans* octameric apoptosome which is formed when the inhibition of CED-4 by CED-9 is disrupted through binding of BH-3-only proteins EGL-1 and CED-13 in the soma and germline, respectively (*Conradt et al., 2016*; *Gartner et al., 2008*). While somatic cell death is blocked by the *ced-9(n1950)* gain-of-function mutation, normal physiological germ cell death is not (*Gumienny et al., 1999*). In keeping with this finding, we found no significant difference in fractional abundance of the mitochondrial deletion in the *ced-9(n1950)* mutant background as compared to a wildtype nuclear genetic background. We posit that in response to mitochondrial genotoxic stress (increased mtDNA$^{uaDf5}$ load), the CED-3 and CSP-1 caspases are activated by the BH-3-only domain protein CED-13, thereby triggering mitochondrial purifying selection through a non-canonical germ cell death pathway, independent of CED-9 and the CED-4 apoptosome (*Figure 6A*).

Mutations that eliminate the function of caspases that act in PCD result in elevated mtDNA$^{uaDf5}$ levels. Two mutations affecting the p15 domain of the CED-3 caspase result in a significantly increased molar ratio of *uaDf5*, highlighting the importance of the p15 domain in mtDNA quality control. It is noteworthy that the two mutations that result in a more substantial effect on mtDNA$^{uaDf5}$ abundance affect CED-3 structure more dramatically: *ced-3(n717)* results in a splicing error, and *ced-3(n1286)* is a nonsense mutation (*Shaham et al., 1999*). In contrast, *ced-3(n2454)*, which results in a subtle (statistically insignificant) increase in mtDNA$^{uaDf5}$, is a substitution predicted to impart a much weaker effect on the protein structure (*Shaham et al., 1999*). The effect of the *ced-3(n718)* allele, which resides in the prodomain of CED-3, suggests that that portion of the protein may act to inhibit mtDNA quality control. Taken together, our results suggest that CED-3 may carry out a specialized activity in mtDNA-activated PCD. We found that a second caspase, CSP-1, which plays a minor role in PCD (*Denning et al., 2013*; *Shaham and Shaham, 1998*), is also required in mtDNA quality control: a *csp-1(−)* knockout mutation results in increased abundance of mtDNA$^{uaDf5}$ and enhances the effect of *ced-3(n717)*, suggesting that CSP-1 may play a larger role in removal of defective mtDNA than it does in other forms of PCD. An exciting possibility is that caspases act in mtDNA quality control via a mechanism that is distinct from their normal action in PCD. Such putative roles for caspases and other mitochondrial factors in non-apoptotic mitochondrial quality control may have been co-opted during metazoan evolution with the innovation of PCD.

Analysis of additional PCD components further implicates a role for germline PCD in mitochondrial purifying selection. These include CED-13, the germline-specific PCD effector, and components of the cell corpse engulfment pathway, which activate PCD likely through a complex feedback mechanism that ensures cells destined to die proceed irreversibly through the process (*Reddien and Horvitz, 2004*; *Hoeppner et al., 2001*). In all cases, mutations in these components result in increased mtDNA$^{uaDf5}$ levels. In vertebrates, mitochondrial reactive oxygen species (mtROS) trigger apoptosis via the intrinsic mitochondrial pathway (*Simon et al., 2000*). Interestingly, elevated mtROS promotes longevity that is in part dependent on the core cell death machinery in *C. elegans*, involving CED-9, CED-4, CED-3, and CED-13, but not EGL-1 (*Yee et al., 2014*). In that study, the authors reported that the protective effect of the cell death machinery on lifespan was independent of PCD in the soma; however, germline cell death was not characterized. Given that CED-13, and not EGL-1, is the predominant BH3-domain protein functioning in the germline (*King et al., 2019*) and that ablation of the germline leads to extended lifespan (*Hsin and Kenyon, 1999*), our findings support the possibility that germline progenitors carrying defective mitochondria selectively undergo PCD, ensuring homeostatic mtDNA copy number and health of progeny.

A striking exception to our findings was seen with mutations that eliminate the function of the proapoptotic regulator CED-4. Neither the *ced-3(n718)* mutation that disrupts the CED-3 CARD domain, which is involved in recruitment to the apoptosome by stabilizing its interaction with CED-4 (*Huang*

et al., 2013; Dorstyn et al., 2018), nor two ced-4(−) mutations, result in increased accumulation of mtDNA$^{uaDf5}$, suggesting non-canonical, CED-4-independent activation of CED-3 in mitochondrial purifying selection. It is possible that the CARD mutation (G65R) in the ced-3(n718) mutant (Shaham et al., 1999) reduces the fraction of CED-3 in complex with the apoptosome, which might release more of the protein for its role in mitochondrial purifying selection. Interestingly, the CARD linker domain has been found to have an inhibitory effect on the pro-caspase-9 zymogen (Yuan et al., 2011). ced-3(n718) could be effectively acting as a gain-of-function allele in the process of purifying selection, reflected by the lower levels of mtDNA$^{uaDf5}$ in this mutant background.

It is noteworthy that CED-4 and its mammalian Apaf1 relatives regulate a variety of cellular functions that are unrelated to their activities in PCD. These include cell growth control influenced by DNA damage, centrosomal function and morphology, neuronal regeneration, and inhibition of viral replication (Zermati et al., 2007; Ferraro et al., 2011; Liu et al., 2006; Wang et al., 2019). In addition, as a result of differential RNA splicing, ced-4 encodes proteins with opposing activities, generating both an activator and a repressor of apoptosis (Chaudhary et al., 1998), which further complicates analysis of its action. Thus, it is conceivable that CED-4 might exert opposing effects on purifying selection, reflecting its pleiotropic activities in development and confounding an unambiguous interpretation of its action in this process.

## IIS and CLK-1 synergistically regulate germline accumulation of mtDNA$^{uaDf5}$ as adults age

We found that most of the increase in the fractional abundance of mtDNA$^{uaDf5}$ as worms age throughout the period of self-fertility (days 1–4) occurs in the germline. However, the relative amount of the defective mtDNA continues to increase in older animals when the germline is no longer actively proliferating (Kimble and Crittenden, 2005), suggesting that mtDNA proliferation also occurs in somatic tissues throughout the aging process. This behavior mirrors the dynamics of mutant mtDNAs observed in other organisms, including human, mouse, rat, and rhesus monkey (Larsson, 2010; Szczepanowska and Trifunovic, 2017; Kujoth et al., 2005). However, in contrast to what is observed over the relatively long lifespan of mammals, the expansion in the burden of defective mitochondrial DNA over the span of a few days in C. elegans is likely the outcome of a need to maintain wildtype levels of mitochondrial DNA that supports the energy demands over the short but intense period of gravidity, during which a single hermaphrodite produces hundreds or thousands of progeny. The replicative advantage of the mtDNA$^{uaDf5}$ deletion may impose a greater replication drive to maintain wildtype mtDNA levels, resulting in 'runaway' expansion of mutant mtDNA. In this scenario, the mechanisms that normally act to eliminate the defective mtDNA would be insufficient to keep up with the increased burden arising from replicative advantage. The expansion the mitochondrial deletion mutation in somatic tissues of old (day 10) post-gravid adults might be an outcome of different dynamics or even an absence of mitochondrial purifying selection mechanisms in the germline. While removal of the germline in worms results in extended lifespan (Mack et al., 2017; Libina et al., 2003), it is not clear whether, or to what extent, this increased lifespan might be attributable to accumulation of mutant mtDNA, since the lack of germline leads to a variety of cellular responses (Lapierre and Hansen, 2012; Antebi, 2013), any of which might lead to lifespan extension.

As in many other organisms, the offspring of older C. elegans mothers show increased embryonic lethality (Andux and Ellis, 2008; Scharf et al., 2021). This age-dependent lethality positively correlates with an increase in accumulation of defective mtDNA. Age-associated changes in the germline include reduced germ cell proliferation, a displaced distal tip cell, germline shrinking, reduction in oocyte production, oocyte clustering, and endomitotic oocytes (Scharf et al., 2021). We observed that gonads of mtDNA$^{uaDf5}$ mutants exhibit accelerated signs of aging (not shown) and smaller brood sizes. Interestingly, long-lived mutants are associated with reduced brood sizes (Antebi, 2007; Hekimi, 2006) and extension in the period of egg production, owing to a decrease in oocyte production, as seen in daf-2(−) mutants in which sporadic embryos are laid for as late as 50 days, a more than 10-fold increase over normal (Gems et al., 1998). This effect is also seen as a consequence of caloric restriction in Eat mutants, which show an egg-laying period that lasts up to 10 days, compared to 3–5 days in the N2 laboratory strain (Pickett and Kornfeld, 2013). One of the targets of pro-longevity cues is suppression of vitellogenin expression, a major maternal energy cost (DePina et al., 2011; Murphy et al., 2003; Perez and Lehner, 2019). While vitellogenin is largely dispensable for

embryogenesis, the accumulation of vitellogenin in post-reproductive animals reaches pathological levels as a consequence of autophagy-dependent degradation of the worm's intestinal tissues, which fuels continued massive production of yolk. The complex interaction between reproductive aging, egg production, transgenerational effects of vitellogenins, and levels of mitochondrial mutation load on life and healthspan are intriguing relationships that demand further study.

Our findings do not reveal whether, or how, aging and accumulation of defective mtDNA are causally linked. However, our findings that steady-state levels of mtDNA$^{uaDf5}$ are lowered in long-lived mutants (*daf-2* (IIS pathway; *Murphy and Hu, 2013*) and *clk-1* mitochondrial function; *Branicky et al., 2000*; *Hekimi and Guarente, 2003*) and that rates of its accumulation are strongly inversely correlated with lifespan extension through independent pathways, suggests that mtDNA purifying selection mechanisms are influenced by aging programs (*Figure 6A*). Further bolstering this potential link is our finding that short-lived mutants (*daf-16* and *aak-2*, both involved in the IIS pathway; *Murphy and Hu, 2013*), show higher steady-state levels of mtDNA$^{uaDf5}$. That this effect is greater in *aak-2* mutants than in *daf-16* mutants suggests that the AAK-2 branch of the IIS pathway influences mtDNA quality control more significantly than does the DAF-16 branch. One of the substrates of AAK-2 is SKN-1/Nrf2, a multifaceted transcription factor with roles in stress response and longevity, and one of its isoforms, SKN-1a, localizes to the mitochondrial surface (*Blackwell et al., 2015*), raising the possibility that AAK-2 might influence clearance of defective mtDNA through SKN-1 action. It will be of interest to assess how defective mtDNA might coordinately trigger quality control and stress-response pathways.

Our observation that IIS pathway components and CLK-1 act synergistically on the mtDNA quality control machinery raises the possibility that these two distinct lifespan-regulating pathways converge on a common system for removal of defective mtDNA, such as a global mitochondrial stress response pathway. Candidates for mediating this removal process include mitochondrial fission/fusion (*Ni et al., 2015*; *Tilokani et al., 2018*; *Youle and van der Bliek, 2012*), mitophagy (*Ashrafi and Schwarz, 2013*; *Youle and Narendra, 2011*; *Twig et al., 2008*), and the UPR$^{MT}$ (*Rolland et al., 2019*; *Münch, 2018*; *Callegari and Dennerlein, 2018*; *Gitschlag et al., 2016*; *Nargund et al., 2012*; *Lin et al., 2016*), all of which are known to act in mtDNA quality control, as well as modulation of the regulatory pathway for PCD, as suggested by our findings. It is possible that along with delaying germ cell replication, aging programs could also slow the replication of mtDNA thereby possibly attenuating the runaway replicative advantage of mtDNA harboring large deletions like *uaDf5* thereby reducing the fractional abundance of *mtDNA$^{uaDf5}$*.

## PCD is uncoupled from aging during intergenerational mitochondrial purifying selection in older mothers

Analysis of newly hatched L1 larvae revealed that the relative load of mtDNA$^{uaDf5}$ is transmitted from mother to offspring, with evidence for intergenerational purifying selection (*Figure 6B*). This finding implies that mtDNA quality control occurs between germline stem cell expansion in the mature female germline and L1 hatching, a developmental period that spans many potential stages at which it might occur, including germline PCD, oocyte maturation, and the entirety of embryogenesis. It is conceivable that this selection process acts at multiple stages throughout this developmental window and that the decreased burden of defective mtDNA in newly hatched L1 larvae reflect the summation of a series of sequentially acting processes that incrementally enrich for healthy mtDNA.

The efficacy of intergenerational removal of mtDNA$^{uaDf5}$ increases in old mothers, including in strains lacking pro-apoptotic regulators. The clearance is particularly precipitous in strains with a very high burden of defective mtDNA as seen in the absence of CED-13, CED-10, and ATFS-1, suggesting a critical threshold beyond which a germline PCD-independent mtDNA quality control process may be triggered in these older mothers (*Figure 6B*). One possible explanation for this observation is that an mtDNA purifying selection mechanism that is typically inhibited by germline PCD might be activated in older mothers. This hypothesized purifying selection mechanism might be triggered by the unique cellular environment associated with aging such as increased organelle or macromolecule damage. Alternatively, the effect might be the result of an age-dependent genetic program.

While our study has uncovered new mechanisms acting in mtDNA purifying selection and its relationship to aging, it is of note that some level of mtDNA$^{uaDf5}$ is maintained in all but the most extreme conditions we have observed (e.g., in the *daf-2(e1391) clk-1(qm30)* double mutant, in which the

defective mtDNA is extirpated). This finding raises the possibility that some degree of heteroplasmy, even with defective mtDNA, is not only tolerated, but may be adaptive by providing a degree of evolutionary plasticity. Cells might purposefully allow for limited heteroplasmy as a way of increasing genetic heterogeneity that might prove evolutionarily advantageous. Such heterogeneity may also be essential to allow mtDNA to co-evolve with changes arising in the nuclear genome. It may be that a dynamic balance between active mtDNA purifying selection, including the mechanisms identified here, and the permissibility of limited heteroplasmy, is modulated according to environmental or physiological demands.

## Materials and methods
### Culturing of nematodes
Nematode strains were maintained on NGM plates as previously described at either 20 or 15°C for the temperature-sensitive strains (*Stiernagle, 2006*). *Supplementary files 13 and 4* provide details of all strains used in this study. Strains without a JR designation were either provided by the CGC which is funded by NIH Office of Research Infrastructure Programs (P40 OD010440) or were obtained from the Mitani lab (strains with a FX designation or JR strains containing alleles with a tm designation were generated from Mitani lab strains) (*Barstead et al., 2012*).

### Population collection by age
Upon retrieval of a stock plate for a given strain, three chunks were taken from the stock plate and placed onto three separate large NGM plates to create three biological replicates (lines). Each of these lines was chunked approximately each generation to fresh large NGM plates (every 3 days if maintained at 20 or 25°C, or every 4 days if maintained at 15°C, being careful to not let the worms starve between chunks). After four generations of chunks, an egg prep was performed on each line (as described previously; *Stiernagle, 2006*) and left to spin in M9 overnight to synchronize the hatched L1s. The next day, each egg prep was plated onto three large seeded plates at an equal density and the worms were left to grow to day 2 adults (second day of egg laying). The day 2 adult worms were egg prepped for synchronization and left to spin in M9 buffer overnight. The next day, each egg prep was plated onto five large NGM plates at equal density. Once the worms reached day 1 of adulthood (first day of egg-laying), one of the plates was used to collect 200 adult worms by picking into 400 µl of lysis buffer, and the remaining adults on the plate were egg prepped for the collection of hatched L1 larvae in 400 µl lysis buffer the following day. The worms on the four remaining plates were transferred to a 40-µm nylon mesh filter in order to separate the adults from the progeny, and the resulting adults were resuspended in M9 and pipetted onto fresh large NGM plates. This process was repeated for the following 3 days (days 2–4 of adulthood). Day 5–10 adults were moved to fresh NGM plates every second day using a 40-µm nylon mesh filter, and the resulting day 10 adults were collected in lysis buffer.

### ddPCR
The worm lysates were incubated at 65°C for 4 hr and then 95°C for 30 min to deactivate the proteinase K. Each lysate was diluted; 100-fold for 200 worm adult population lyses, 2-fold for 200 worm L1 population lyses, and 25-fold for individual adult lyses. 2 µl of the diluted lysate was then added to 23 µl of the ddPCR reaction mixture, which contained a primer/probe mixture and the ddPCR probe supermix with no dUTP. The primers used were:

> WTF: 5′-GAGGGCCAACTATTGTTAC-3′
> WTR: 5′-TGGAACAATATGAACTGGC-3′
> UADF5F: 5′-CAACTTTAATTAGCGGTATCG-3′
> UADF5R: 5′-TTCTACAGTGCATTGACCTA-3′

The probes used were:

> WT: 5′-HEX-TTGCCGTGAGCTATTCTAGTTATTG-Iowa Black FQ-3′
> UADF5: 5′-FAM-CCATCCGTGCTAGAAGACAAAG-Iowa Black FQ-3′

The ddPCR reactions were put on the Bio-Rad droplet generator and the resulting droplet-containing ddPCR mixtures were run on a Bio-Rad thermocycler with the following cycle parameters, with a ramp rate of 2°C/s for each step:

1. 95°C for 5 min
2. 95°C for 30 s
3. 60°C for 2 min
4. Repeat steps 2 and 3 40×
5. 4°C for 5 min
6. 90°C for 5 min

After thermocycling, the ddPCR reaction plate was transferred to the Bio-Rad droplet reader and the Quantasoft software was used to calculate the concentration of mtDNA$^{uaDf5}$ (FAM positive droplets) and mtDNA$^{WT}$ (HEX positive droplets) in each well.

## Lifespan analysis

Confluent large plates were egg prepped and left to spin in M9 overnight for synchronization. The hatched L1s were plated onto large thick plates and allowed to grow to day 2 adults before being egg prepped a second time and left to spin in M9 overnight. The next morning, referred to as day 1 for lifespan determination, L1s were singled out onto small plates. Once the worms started laying eggs, they were transferred each day to a fresh small plate until egg laying ceased, after which the worms remained on the same plate unless bacterial contamination required transfer to a fresh plate. Worms were considered dead if there was no movement after being lightly prodded with a worm pick. Worms that died due to desiccation on the side of the plate were excluded from analysis.

## Brood size and embryonic lethality analysis

Confluent large plates were egg prepped and left to spin in M9 overnight for synchronization. The hatched L1s were plated onto large thick plates and allowed to grow to day 2 adults before being egg prepped a second time and left to spin in M9 overnight. The next morning, L1s were singled out onto small plates. Once the worms started laying eggs, they were transferred each day to a fresh small plate until egg laying ceased. The day after transfer to a fresh plate, unhatched embryos and hatched larvae on the plate from the previous day were counted. This was done for each of the days of laying and the total of unhatched embryos and hatched larvae from all plates from a single worm were tabulated to determine total brood size. To determine embryonic lethality, the total number of unhatched embryos was divided by the total brood size. Worms that died due to desiccation on the side of the plate were excluded from analysis.

## Developmental time course analysis

Confluent large plates were egg prepped and left to spin in M9 overnight for synchronization. The hatched L1s were plated onto large thick plates and allowed to grow to day 2 adults before being egg prepped a second time and left to spin in M9 overnight. The next morning, L1s were singled out onto small plates. The stage of the worms was assayed every 12 hr for the first 72 hr after plating. For determining the stage at 60 hr, L4 worms were divided up into three subgroups based on morphology: young-L4, mid-L4, and late-L4; otherwise, all other staged worms were not divided up into subgroups. Worms that died due to desiccation on the side of the plate were excluded from analysis.

## Genotyping the w47 allele

DNA was collected from reference strain (N2) and two uaDf5-containing strains, LB138 and JR3630. Mitochondrial DNA was extracted and the DNA libraries were prepared using Nextera Kit and then sequenced using an Illumina NextSeq500. Prior to alignment, reads from fastq files were trimmed using Trimmomatic. Trimmed, pair-end reads (2 × 150) were then mapped to the *C. elegans* assembly reference sequence WBcel235 using Burrows-Wheeler Aligner (BWA) (*Li and Durbin, 2009*). Picard Tools (http://broadinstitute.github.io/picard/) was used to mark duplicate reads, and SAMtools (*Li et al., 2009*) was used to merge, index, and create pile-up format. VarScan (*Koboldt et al., 2009*) was used to call variants, and only variants with minimum coverage of 100 and a minimum variant frequency call of 0.01 were considered for analysis.

## Statistical analysis

Summary statistics, analysis of variance, Mann–Whitney tests, and linear regression were calculated using R v3.4.1. The details of the statistical tests are reported in the figure legends.

## Acknowledgements

We would like to thank the members of the Rothman lab for their support. We thank Jen Smith and the BNL at UCSB for providing excellent facilities which were necessary for this work. We thank Kyle Ploense for his help in statistical analysis. Worm strains used in this work were provided by the Mitani lab, as well as the *Caenorhabditis* Genetics Center (CGC), which is funded by NIH Office of Research Infrastructure Programs Grant P40 OD010440. Funding. This work was supported by the grants from the National Institutes of Health (#R01HD082347, #R01HD081266, #R01GM143771, and #R21AG068915).

## Additional information

### Funding

| Funder | Grant reference number | Author |
|---|---|---|
| National Institutes of Health | R01HD082347 | Sagen Flowers<br>Joel H Rothman |
| National Institutes of Health | R01HD081266 | Sagen Flowers<br>Rushali Kothari<br>Yamila N Torres Cleuren<br>Melissa R Alcorn<br>Chee Kiang Ewe<br>Geneva Alok<br>Samantha L Fiallo<br>Pradeep M Joshi<br>Joel H Rothman |
| National Institutes of Health | R01GM143771 | Chee Kiang Ewe<br>Geneva Alok<br>Samantha L Fiallo<br>Pradeep M Joshi<br>Joel H Rothman |
| National Institutes of Health | R21AG068915 | Sagen Flowers<br>Rushali Kothari<br>Joel H Rothman |

The funders had no role in study design, data collection, and interpretation, or the decision to submit the work for publication.

### Author contributions

Sagen Flowers, Conceptualization, Formal analysis, Funding acquisition, Validation, Investigation, Visualization, Methodology, Writing - original draft, Writing - review and editing; Rushali Kothari, Yamila N Torres Cleuren, Investigation, Writing - review and editing; Melissa R Alcorn, Formal analysis, Writing - review and editing; Chee Kiang Ewe, Geneva Alok, Samantha L Fiallo, Writing - review and editing; Pradeep M Joshi, Conceptualization, Supervision, Investigation, Writing - review and editing; Joel H Rothman, Conceptualization, Supervision, Funding acquisition, Writing - review and editing

### Author ORCIDs

Sagen Flowers http://orcid.org/0000-0002-7818-2188
Melissa R Alcorn http://orcid.org/0000-0001-6284-3255
Chee Kiang Ewe http://orcid.org/0000-0003-1973-1308
Pradeep M Joshi http://orcid.org/0000-0002-4220-0559
Joel H Rothman http://orcid.org/0000-0002-6844-1377

### Decision letter and Author response

Decision letter https://doi.org/10.7554/eLife.79725.sa1

Author response https://doi.org/10.7554/eLife.79725.sa2

## Additional files

### Supplementary files

• Supplementary file 1. A summary of all mutants analyzed in the PCD pathway, including their known homologs, whether they are part of the core PCD machinery, if they are pro- or anti-apoptotic, whether they are mitochondrial proteins, and molecular details of the alleles analyzed.

• Supplementary file 2. Wildtype and mutant mtDNA levels in the different cell death and aging pathway mutants. (mean of 3 replicates ± standard deviation).

• Supplementary file 3. A summary of all lifespan mutants analyzed, including their known homologs, cellular pathways they are known to act in, whether the mutant extends or reduces lifespan, and molecular details of the alleles analyzed.

• Supplementary file 4. *C. elegans* strains used in this study.

• MDAR checklist

### Data availability

All data generated or analyzed during this study are included in this article and accompanied supplementary materials. The raw reads of the sequenced N2, LB138, and JR3688 genomes have been deposited at the NCBI SRA under the study accession number PRJNA836592.

The following dataset was generated:

| Author(s) | Year | Dataset title | Dataset URL | Database and Identifier |
|---|---|---|---|---|
| Flowers S | 2022 | *C. elegans* uaDf5 sequencing | https://www.ncbi.nlm.nih.gov/sra/PRJNA836592 | NCBI Sequence Read Archive, PRJNA836592 |

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
