## [Editor Report]

This valuable work suggests a novel mechanism of purifying selection by which programmed cell death contributes to the selective removal of mtDNA deletion mutations in *C. elegans*. The authors demonstrate with convincing evidence that mtDNA deletion is more abundant in the offspring of older mothers and that activators for programmed cell death are required to suppress the accumulation of defective mtDNA in the germline of *C. elegans*. Because of the likely central role of mtDNA deletions in aging and age-dependent diseases, this work will be of interest to scientists in the field of mitochondrial biology as well as aging.

---

## [Decision Letter]

**Decision letter after peer review:**

Thank you for submitting your article "Regulation of defective mitochondrial DNA accumulation and transmission in *C. elegans* by the programmed cell death and aging pathways" for consideration by *eLife*. Your article has been reviewed by 2 peer reviewers, and the evaluation has been overseen by a Reviewing Editor and Benoît Kornmann as the Senior Editor. The reviewers have opted to remain anonymous.

The reviewers were generally enthusiastic about this manuscript and the novelty of the proposed mechanism. There was agreement that the work will be of wide interest but the reviewers also pointed out limitations and suggested some further work. The Reviewing Editor has collated these into the following numbered list of questions and essential revisions:

1. As noted in the public review, the lack of readouts on the copy numbers of mtDNA (mutant and WT) makes it difficult to pinpoint the overall effects of molecular KO of PCD machinery as well as perturbations to aging pathways on mtDNA quality control in the germline. I strongly recommend having data of these copy numbers or their proxies, which hopefully can be extracted from ddPCR data and can be compared among different strains. With these data, a clearer and more definitive view of the regulation of mtDNA quality control in the germline will emerge.

2. Based on the mutant-to-WT ratio data, another explanation for the role of PCD cannot be rejected. In this alternative explanation, stochastic cell divisions and mitochondrial dynamics (biogenesis, degradation, fusion-fission) produce cells with different mutant mtDNA abundances. Cells harboring a high level of the mutant are generated randomly which are then removed via PCD. A dysfunctional PCD will lead to an increased steady-state mutant-to-WT ratio as reported, since cells with a high mtDNA mutant burden remain in the population. But, an overall increase of PCD without a concomitant increase in the rate of generation of cells with a high mutant burden will presumably lead to a general increase in the removal of cells without regard of their mutant burden. This will keep the steady-state mutant-to-WT ratio in the germline unchanged, as reported. Unlike what is stated in Finding #2 in the Discussion, PCD does NOT have any non-canonical / non-apoptotic role in the selection of cells to be removed.

3. Are there data to support the selective removal of cells with a high burden of mutant mtDNA uaDf5? At what ratio of mutant-to-WT mtDNA do we expect to see this preferential removal?

4. Ideally, data on the apoptotic activity should also be provided, especially in the connection to aging and aging pathways. It is rather surprising that the connection between aging and PCD is not explored more deeply in this study. How does aging affect PCD in *C. elegans* carrying mutant mtDNA uaDf5, and does this effect explain the age-related increase of mutant-to-WT mtDNA ratio?

5. The authors should discuss how the egg-laying rate of long-living *C. elegans* over different ages might play a role in the overt increase of mutant-to-WT ratio? Many of these long-living mutants have an altered (delayed and/or extended) egg-laying period. How might this affect the mtDNA deletion dynamics?

6. The continued increase of the fractional abundance of mutant mtDNA uaDf5 beyond day 4 is quite interesting and surprising, especially considering the increase spanned only 6 days (day 4 to day 10). This short span is starkly different from the time span for mtDNA mutant accumulation in organisms like human, mouse, rat, and rhesus monkey (possibly over months to years). Measurements of WT and mutant copy numbers may reveal what underlies such observation.

One difference here is the possibly initial burden of mutation as the expansion of intrinsic (initially extremely low copy number) mtDNA deletions has been suggested to be extremely slow/rare in *C. elegans*.

7. It was surprising to learn that uaDf5 maintenance does not depend on complementing a mutation in other copies of mtDNA. An alternative explanation might be that uaDf5 has a replication advantage over wt mtDNA. This should be addressed.

8. Related to 7, although less likely, it is also possible that long-lived mutants have slower and thus more tightly controlled mtDNA replication, allowing for preferential replication of wt mtDNA.

9. Potential effects of the long-lived mutants on mitophagy and its role in the clearance of defective mitochondria is not addressed, even though that could explain the observed effects of the long-lived mutants on uaDf5 clearance.

10. If a bottleneck at a late stage is responsible for culling the excess mutant mtDNA in the PCD mutants, then the effects would be stochastic (with a large SD) rather than yielding consistent numbers as are observed with this mutant. This prediction could potentially be addressed by a simplified stochastic model of mtDNA turnover?

11. It would be interesting to know whether cps-6/endoG also contributes to the removal of mutant mtDNA independent of PCD.

12. The last part of the Results section describing inter-generation removal is written in a very confusing manner. It would help if the inter-generational removal in Figure 3C is combined with Figure 5.

*Reviewer #1 (Recommendations for the authors):*

It was surprising to learn that uaDf5 maintenance does not depend on complementing a mutation in other copies of mtDNA. An alternative explanation might be that uaDf5 has a replication advantage over wt mtDNA. This should be addressed.

Potential effects of the long-lived mutants on mitophagy and its role in clearance of defective mitochondria is not addressed, even though that could explain the observed effects of the long-lived mutants on uaDf5 clearance.

Although less likely, it is also possible that long-lived mutants have slower and thus more tightly controlled mtDNA replication, allowing for preferential replication of wt mtDNA.

If a bottleneck at a late stage is responsible for culling the excess mutant mtDNA in the PCD mutants, then the effects would be stochastic (with a large SD) rather than yielding consistent numbers as are observed with this mutant.

It would be interesting to know whether cps-6/endoG also contributes to removal of mutant mtDNA independent of PCD.

The last part of the Results section describing intergeneration removal is written in a very confusing manner. It would help if the intergenerational removal in Figure 3C is combined with Figure 5.

*Reviewer #2 (Recommendations for the authors):*

1. As noted in the public review, the lack of readouts on the copy numbers of mtDNA (mutant and WT) makes it difficult to pinpoint the overall effects of molecular KO of PCD machinery as well as perturbations to aging pathways on mtDNA quality control in the germline. I strongly recommend having data of these copy numbers or their proxies, which hopefully can be extracted from ddPCR data and can be compared among different strains. With these data, a clearer and more definitive view on the regulation of mtDNA quality control in the germline will emerge.

2. Based on the mutant-to-WT ratio data, another explanation for the role of PCD cannot be rejected. In this alternative explanation, stochastic cell divisions and mitochondrial dynamics (biogenesis, degradation, fusion-fission) produce cells with different mutant mtDNA abundances. Cells harboring a high level of mutant are generated randomly which are then removed via PCD. A dysfunctional PCD will lead to increased steady-state mutant-to-WT ratio as reported, since cells with a high mtDNA mutant burden remain in the population. But, an overall increase of PCD without a concomitant increase in the rate of generation of cells with a high mutant burden will presumably lead to a general increase in the removal of cells without regards of their mutant burden. This will keep the steady-state mutant-to-WT ratio in the germline unchanged, as reported. Unlike what is stated in Finding #2 in Discussion, PCD does NOT have any non-canonical / non-apoptotic role in the selection of cells to be removed.

3. Are there data to support the selective removal of cells with high burden of mutant mtDNA uaDf5? At what ratio of mutant-to-WT mtDNA do we expect to see this preferential removal?

4. Ideally, data of apoptotic activity should also be provided, especially in the connection to aging and aging pathways. It is rather surprising that the connection between aging and PCD is not explored more deeply in this study. How does aging affect PCD in *C. elegans* carrying mutant mtDNA uaDf5, and does this effect explain the age-related increase of mutant-to-WT mtDNA ratio?

5. Related to the claimed accumulation of mtDNA mutant during aging, the increased ratio of mutant-to-WT could be explained differently. In this alternative explanation, oocytes with lower mutant-to-WT ratio are preferentially laid before those with higher mutant-to-WT ratio. Over time, the oocytes remaining in mothers will have an increasing mutant-to-WT ratio, and thus the oocytes will follow the same trend. We may not be able to reject this alternative explanation until we have data on mtDNA copy numbers in the germline over different ages. Note that an increase in the ratio of mutant-to-WT with age cannot be interpreted immediately as an accumulation of mutant mtDNA-a term that implies a preferential production / replication of mutant mtDNA over WT mtDNA.

6. How does the egg-laying rate of long-living *C. elegans* over different ages play a role in the overt increase of mutant-to-WT ratio? Many of these long-living mutants have altered (delayed and/or extended) egg-laying period.

7. The continued increase of fractional abundance of mutant mtDNA uaDf5 beyond day 4 is quite interesting and surprising, especially considering the increase spanned only 6 days (day 4 to day 10). This short span is starkly different from the time span for mtDNA mutant accumulation in organisms like human, mouse, rat and rhesus monkey (possibly over months to years). Measurements of WT and mutant copy numbers may reveal what underlies such observation.

---

## [Author Response]

Essential revisions:The reviewers were generally enthusiastic about this manuscript and the novelty of the proposed mechanism. There was agreement that the work will be of wide interest but the reviewers also pointed out limitations and suggested some further work. The Reviewing Editor has collated these into the following numbered list of questions and essential revisions:1. As noted in the public review, the lack of readouts on the copy numbers of mtDNA (mutant and WT) makes it difficult to pinpoint the overall effects of molecular KO of PCD machinery as well as perturbations to aging pathways on mtDNA quality control in the germline. I strongly recommend having data of these copy numbers or their proxies, which hopefully can be extracted from ddPCR data and can be compared among different strains. With these data, a clearer and more definitive view of the regulation of mtDNA quality control in the germline will emerge.

We agree that these copy number data are very important and should have been included in the original manuscript. This appears to be the major concern noted in the reviews. In response to this concern, we have added the requested data throughout the revision, and modified the figures accordingly to incorporate these new results (modified Figures 2, 3, and 4). These data indicate that, in most cases, there is a variable and at most modest increase in the overall copy number of the normal mtDNA (mtDNA*^WT^*) across the various genetic backgrounds, as we describe in the revised text and the new Supplementary Table 1. These data do not alter, but provide more definitive support for, the key conclusions of the paper.

2. Based on the mutant-to-WT ratio data, another explanation for the role of PCD cannot be rejected. In this alternative explanation, stochastic cell divisions and mitochondrial dynamics (biogenesis, degradation, fusion-fission) produce cells with different mutant mtDNA abundances. Cells harboring a high level of the mutant are generated randomly which are then removed via PCD. A dysfunctional PCD will lead to an increased steady-state mutant-to-WT ratio as reported, since cells with a high mtDNA mutant burden remain in the population. But, an overall increase of PCD without a concomitant increase in the rate of generation of cells with a high mutant burden will presumably lead to a general increase in the removal of cells without regard of their mutant burden. This will keep the steady-state mutant-to-WT ratio in the germline unchanged, as reported. Unlike what is stated in Finding #2 in the Discussion, PCD does NOT have any non-canonical / non-apoptotic role in the selection of cells to be removed.

We agree with this explanation and, in fact, believe that it is a likely mechanism explaining the requirement for the cell death machinery. We have revised the discussion to clarify that mitochondrial stress may indeed activate germ cell death. As we explain, such a stress pathway appears to be independent of CED-4, in contrast to physiological and genotoxic stress-induced germ cell death, both of which are dependent on both CED-3 and CED-4. We therefore propose that this process may act through a non-canonical cell death pathway (lines 459, 513-520), an exciting possibility that will be pursued in follow-up studies.

3. Are there data to support the selective removal of cells with a high burden of mutant mtDNA uaDf5? At what ratio of mutant-to-WT mtDNA do we expect to see this preferential removal?

These are fascinating and important questions that will require extensive future investigations. The correlative information indicated in Figure 2A does indeed suggest the possibility that a threshold event occurs. However, investigating whether a critical mutant-to-WT ratio results in cell removal will require quantitatively analyzing mutant and WT mtDNA levels in *individual* cells dynamically over time and assessing the fates of cells (e.g., death or not) with different ratios. Such experiments require development of advanced new techniques. While these complex questions are certainly of high interest, they represent an entirely new methodological approach and are well beyond the scope of the current paper (see footnote regarding additional studies below^1^).

4. Ideally, data on the apoptotic activity should also be provided, especially in the connection to aging and aging pathways. It is rather surprising that the connection between aging and PCD is not explored more deeply in this study. How does aging affect PCD in *C. elegans* carrying mutant mtDNA uaDf5, and does this effect explain the age-related increase of mutant-to-WT mtDNA ratio?

These are certainly important and interesting questions that we are in the process of investigating. However, the paper is already quite extensive (please note that, as reviewer #2, stated, the paper is a “comprehensive exploration of the role of key molecules”) and these questions require new lines of investigation that are well beyond the scope of the paper. The findings from such studies, which will require additional person-years, would not change the fundamental conclusions of the current study and therefore are appropriate for a subsequent.

5. The authors should discuss how the egg-laying rate of long-living *C. elegans* over different ages might play a role in the overt increase of mutant-to-WT ratio? Many of these long-living mutants have an altered (delayed and/or extended) egg-laying period. How might this affect the mtDNA deletion dynamics?

This issue has now been discussed in the revision (lines 526-614).

6. The continued increase of the fractional abundance of mutant mtDNA uaDf5 beyond day 4 is quite interesting and surprising, especially considering the increase spanned only 6 days (day 4 to day 10). This short span is starkly different from the time span for mtDNA mutant accumulation in organisms like human, mouse, rat, and rhesus monkey (possibly over months to years). Measurements of WT and mutant copy numbers may reveal what underlies such observation.One difference here is the possibly initial burden of mutation as the expansion of intrinsic (initially extremely low copy number) mtDNA deletions has been suggested to be extremely slow/rare in *C. elegans*.

We agree with the reviewer that these are interesting results. These data requested by the reviewer have now been incorporated into the revised manuscript and in Figure 3. The results suggest that there is indeed some progressive expansion of mtDNA over several days of adulthood. It is therefore possible that replicative advantage of the *uaDf5-*bearing mtDNA during this expansion phase might account for the increased burden of the defective mtDNA. In such an event, the mechanisms that normally act to eliminate the defective mtDNA (including PCD, as described here, and possibly mitophagy) must be insufficient to keep up with the increased burden. In the revised manuscript, we now discuss these new data and important issues in the Results (lines 321-324) and the Discussion (lines 581-591).

7. It was surprising to learn that uaDf5 maintenance does not depend on complementing a mutation in other copies of mtDNA. An alternative explanation might be that uaDf5 has a replication advantage over wt mtDNA. This should be addressed.

We fully agree; in fact, replicative advantage is exactly the mechanism that we believe most likely explains the stable heteroplasmy of mtDNA*^uaDf5^*. We regret that we did not sufficiently emphasize this point in the original manuscript and have now explicitly included it in the revision (lines 476483).

8. Related to 7, although less likely, it is also possible that long-lived mutants have slower and thus more tightly controlled mtDNA replication, allowing for preferential replication of wt mtDNA.

As noted in the response to comment #6 above, there is an increase in total mtDNA with age and replicative advantage of mtDNA*^uaDf5^* during this expansion might indeed result in insufficient removal of the defective DNA. Thus, if the long-lived mutants have slower rates of replication, the potential replicative advantage of the defective mtDNA might be attenuated, accounting for at least some of the effect we have observed. However, while this might provide a partial explanation for our findings, the fractional representation of mtDNA*^uaDf5^* is dramatically decreased in the longlived mutants, suggesting that other processes are also likely to contribute to the effect. We have now included a passage that discusses these possibilities in the revised Discussion [635-638].

9. Potential effects of the long-lived mutants on mitophagy and its role in the clearance of defective mitochondria is not addressed, even though that could explain the observed effects of the long-lived mutants on uaDf5 clearance.

In the revised Discussion, we now discuss the possibility that the aging program might affect mitophagy.

10. If a bottleneck at a late stage is responsible for culling the excess mutant mtDNA in the PCD mutants, then the effects would be stochastic (with a large SD) rather than yielding consistent numbers as are observed with this mutant. This prediction could potentially be addressed by a simplified stochastic model of mtDNA turnover?

As noted by the reviewers, the effects do not show the large variation that would be consistent with such a possibility. While it would be of some interest to initiate stochastic modeling, our results do not suggest that such a late bottleneck occurs on the basis of this low fluctuation and such modeling is not essential for any of the conclusions presented in the paper. As such, this direction would not be of sufficiently high priority to hold off publication until it has been done. Further, such a modeling effort would require an entirely new direction, including bringing additional expertise to the project, which would pose the challenge of many person-months, if not person-years (see also footnote regarding additional studies below^1^).

11. It would be interesting to know whether cps-6/endoG also contributes to the removal of mutant mtDNA independent of PCD.

This manuscript already reports two different mechanisms that modulate loss of defective mtDNA (the PCD and aging pathways). There are many potential additional contributors to the processes that remove defective mtDNAs and we agree that CPS-6/endoG may be one of them. However, while CPS-6 is an interesting candidate, it is not a standout priority among many such potential candidates. While testing CPS-6, and many other possible candidates, is of interest for future studies, these additional experiments are not required to support any of the major findings or claims in the paper. In future studies, we and other labs will be testing many other candidates. While identifying yet more players in the process will be interesting and informative in future studies, the paper already presents a substantial amount of data and findings and we believe that the results and conclusions that we present stand strongly on their own. (See also footnote regarding additional studies below^1^.)

12. The last part of the Results section describing inter-generation removal is written in a very confusing manner. It would help if the inter-generational removal in Figure 3C is combined with Figure 5.

We regret the confusing text and have revised this section to improve the clarity (lines 440 -445) regarding inter-generational removal.

Moving Figure 3C to later in the paper is problematic because it is important at that earlier stage in the text to show the Day 1-3 data demonstrating the intergenerational removal. We believe our revisions to the last section of the Results now clarify the text that led to confusion and do not require moving Figure 3C.